# Paper and Other Fibrous Materials—A Complete Platform for Biosensing Applications

**DOI:** 10.3390/bios11050128

**Published:** 2021-04-21

**Authors:** Domingo R. Flores-Hernandez, Vivian J. Santamaria-Garcia, Elda M. Melchor-Martínez, Juan Eduardo Sosa-Hernández, Roberto Parra-Saldívar, Jaime Bonilla-Rios

**Affiliations:** Escuela de Ingeniería y Ciencias, Tecnologico de Monterrey, Avenida Eugenio Garza Sada 2501, Monterrey 64849, NL, Mexico; A00819488@itesm.mx (D.R.F.-H.); A00813333@itesm.mx (V.J.S.-G.); elda.melchor@tec.mx (E.M.M.-M.); eduardo.sosa@tec.mx (J.E.S.-H.); r.parra@tec.mx (R.P.-S.)

**Keywords:** paper-based analytical device, paper-based biosensors, electrospun nanofibers for biosensing applications, point-of-care

## Abstract

Paper-based analytical devices (PADs) and Electrospun Fiber-Based Biosensors (EFBs) have aroused the interest of the academy and industry due to their affordability, sensitivity, ease of use, robustness, being equipment-free, and deliverability to end-users. These features make them suitable to face the need for point-of-care (POC) diagnostics, monitoring, environmental, and quality food control applications. Our work introduces new and experienced researchers in the field to a practical guide for fibrous-based biosensors fabrication with insight into the chemical and physical interaction of fibrous materials with a wide variety of materials for functionalization and biofunctionalization purposes. This research also allows readers to compare classical and novel materials, fabrication techniques, immobilization methods, signal transduction, and readout. Moreover, the examined classical and alternative mathematical models provide a powerful tool for bioanalytical device designing for the multiple steps required in biosensing platforms. Finally, we aimed this research to comprise the current state of PADs and EFBs research and their future direction to offer the reader a full insight on this topic.

## 1. Introduction

Noncommunicable diseases, such as tuberculosis, diarrhea, diabetes, lung cancer, and communicable diseases, like lower respiratory infections, are on the top ten of the leading causes of death worldwide [1]. Several factors, such as demographic, socioeconomic, or geographic, create relevant inequalities between developing and developed countries in health diagnostics, monitoring and treatment [2]. Consequently, the death rate is substantially higher in Africa, Southeast Asia and the eastern Mediterranean than in the Americas or the European regions [3]. 

In this regard, the World Health Organization (WHO) has enlisted a set of criteria in search of an ideal suite of technologies useful at the point-of-care (POC), particularly where resources are limited. The WHO ASSURED ( Affordable, Sensitive, Specific, User-friendly, Rapid and robust, Equipment-free, and Deliverable to end-users) criteria leads the technological development of POC to be a complete platform for biosensing applications [4]. In recent years, paper-based analytical devices paper-based analytical devices (PADs) have transformed health, and environmental applications for POC approaches since its conception in 2007 by the pioneering research done by George Whitesides research group [5,6]. PADs usually comprise microfluidic, chemical, biochemical, and nano-composed components into paper or other fibrous substrates such as electrospun fibers. Since electrospun nanofibers materials are being extensively studied for biosensing applications [7], we defined a new category of fibrous biosensors called Electrospun Fiber-Based Biosensors (EFBs). Even though PADs and EFBs are intrinsically affordable, equipment-free, and deliverable to end-users, researchers still struggle to accomplish all the required criteria to produce devices capable of early diagnosis to enable immediate medical decisions making [8].

Therefore, the physicochemical properties that confer the paper the ability to be turned into a fully integrated solution for biosensing applications are reviewed, as well as the classical and alternative models that describe the microfluidic control, diffusivity, and deposition of materials and biological moieties. Moreover, we reviewed the current trends in PADs and EFBs fabrication to ensure the ASSURED criteria, from the selection or production of the paper, bio immobilization approaches, surface modification techniques of paper, to the readout techniques, performance, and field of application. It is worth mentioning that traditionally, the paper is made of cellulose fibers. Still, we can also find different types of paper materials or combinations of materials different to cellulose but preserving its fibrous nature in these devices. Therefore, the paper will be referred to indistinctly as any woven or non-woven fibrous substrate in this work

## 2. Paper and Fibrous Materials as a Complete Platform for Biosensors

Several factors must be addressed when developing biosensors, such as fluidics design (handling of the sample, reduction of sample consumption, optimization of analyte transport, and reduction in detection time.), transduction signal generation (amplification of the signal, decrease of noise, and discriminative readout.), surface immobilization chemistry (high immobilization rate and specific and selective binding [9]. In this regard, the use of paper for biosensing applications is being exploited for diverse aspects—it is low-cost (1), versatile (2), accessible (3), biocompatible (4), chemically and physically modifiable (5), independent from auxiliary equipment for pumping due to its inherent wicking nature (6), and easy to be disposed of (7) [10] which let PADs and EFBs to address the several basic diagnostic operations involved in the bioanalytical assays such as sample collection, sample handling, washing, incubation, filtering, pumping fluids and mixing; see Figure 1 [10,11,12,13].

However, in order to build robust and rapid PADs and EFBs, several factors should be considered, such as the specific material selection and the size and arrangement of the fibers; which, combined with the proper manufacturing process, can provide a customized paper to achieve the physical and chemical properties for multiple purposes in a single device [22]. This section will discuss the physical and chemical properties that are pivotal for the paper to perform the desired functions.

### 2.1. Physical and Chemical Properties 

The fiber mat is usually the main component of a fibrous-based biosensor; thus, its properties will determine the ability to control the fluidic transport, separation, and immobilization of substances in the device [23]. Due to the fibrous nature of these materials, physical parameters are directly related to their constituent fibers. The capillary flow rate, surface area, porosity, permeability, and wettability, are determined by the length, diameter, density, and arrangement of the fibers [13,22,24,25].

Capillary flow rate has one of the highest impact parameters for the analytical performance of these fibrous-based biosensors; the wicking action within the paper substrate is caused by the combination of cohesive and adhesive forces. Cohesive forces, for example, surface tension, results from intermolecular attraction between the fluid at the liquid-air interface [10]. In contrast, adhesive forces result from the intermolecular attraction at the liquid-fiber interface [26].

On the other hand, the chemical properties rely on the availability of active functional groups to immobilize a wide range of materials or biological molecules [26]. Thus, material selection is mostly dictated by the paper’s chemical properties. For the case of conventional paper, manufacturing techniques might reduce its chemical reactivity due to the contamination by reagents required for the fabrication process itself [27]. Therefore, strategies to introduce additional reactive species to fiber surfaces, for example, carboxyl or amine groups, are continuously explored to obtain better interactions with paper [22]. Conversely, paper-like nanofibrous materials can be produced with reactive functional groups by electrospinning, a technique where a polymer solution or melt is subjected to an electrostatic field to produced polymer fibers with nanometer-scale diameters [28]. The introduction of functional groups to the polymer solution before electrospinning eliminates further chemical modifications but increases the production complexity [29]. Once the desired electrospun paper is produced, it can be manipulated and integrated into a sensing platform in the same way as the typical paper is [7]. A more detailed explanation of the electrospinning process is presented in Section 4.1.2, Electrospun mats.

### 2.2. Functions of Fibrous Materials in PADs and EFBs

In PADs and EFBs, the fibrous mat mainly plays three functions: (1) To act as a matrix to immobilize the biorecognition elements; that is, to provide an anchor for enzymes, antibodies, cells, or DNA/RNA probes [30,31,32,33,34]; (2) to act as a substrate to contain/store functionalization agents (e.g., nanostructured materials); and (3) to provide the medium for fluid transport, see Figure 2 [12,35,36,37,38].

A biosensor may use recognition elements such as enzymes, antibodies, DNA/RNA probes, whole cells, and even an artificial version of them [24]. There are various approaches to perform the bio-attachment and can be generally classified as (1) Physical methods, where the biomolecule is confined into the paper surface only by weak forces such as Van der Waals, electrostatic, and hydrogen and hydrophobic interactions; (2) Chemical methods, where the surface is chemically modified in order to create a covalent bond with the biomolecule; and (3) Biological methods where the biomolecule is attached to paper due to biochemical affinity [39,40,41].

The inherently large surface-to-volume ratio of the fibrous materials makes them a suitable substrate to support functionalizing materials [41]. In this regard, nanomaterials, such as carbon inks, carbon nanotubes (CNTs), metal nanoparticles (NPs) (Au, Cu, Pd, Co, Ag, or Pt), oxide NPs (Fe_3_O_4_, TiO_2_ or ZnO) and a wide variety of polymers, have been incorporated to biosensors [42]. For example, conductive materials such as carbon inks, CNTs, or metal NPs are the proper approach for fabricating electrodes for electrochemical biosensors to improve the sensitivity and speed of response [12]. Similarly, the optical detection in PADs and EFBs requires materials to help the multiplexity detection or enhance the response to light upon a specific antigen. For this purpose, AuNPs are widely used due to their color shift on the surface plasmon band [26]. For colorimetric detection, chitosan has been widely used to immobilize biological elements increasing the color shift [43,44,45,46]. Moreover, combining two or more of these materials can show great potential to enhance sensitivity and selectivity [38]. A popular example is a combination of carbon inks with Prussian blue NPs to produce redox reactions for electrochemical detection [30,35,47,48].

The fibrous nature of paper confers the proper structure for fluids to be conducted by capillarity, that is, without external pumping equipment [13]; simultaneously, the network structure provides the ability to filtrate/separate substances according to its mean pore size. In a paper sheet, unless having previously defined hydrophobic barriers, fluids are transported all over it [43]. The fabrication of these barriers is an approach for flow control and gives rise to microfluidic channels in the paper. Generally, the available techniques for this purpose involve physical blocking of pores, physical surface deposition of hydrophobic reagents (e.g., wax), or chemical surface modification of paper [37].

### 2.3. Mechanisms and Modeling

Understanding the fluid dynamics in paper-based microfluidic systems is crucial for designing the multiple operations involved in biosensing applications. Into the porous matrix formed by hydrophilic cellulose, nitrocellulose fibers, or synthetic non-woven polymer matts, natural and spontaneous imbibition of the liquid is present due to capillary forces [13].

Classical models have been proposed to represent the active capillary action, which is the driving force to move fluids in paper-like materials. In bulk conditions, this force is stable at the interface between water and air. On the other hand, when paper fibers are present, two forces appear; cohesion (air-liquid) and adhesion (liquid-solid) [10]. This phenomenon depends on physical properties such as porous size, material, structure, geometry, and permeability combined with the liquid physic characteristics. The fluid transport is usually segregated into two. The first one is the wet-out process that follows the Lucas–Washburn (L–W) model, Equation (1), and the second fully wetted flow that follows Darcy’s law, Equation (6) [49].

The classical L–W equation has been extensively modified to explain the capillary-driven flow’s fundamental porous systems mechanisms. The L–W model assumes several ideal conditions such as incompressible Newtonian fluids, neglects the effect of gravity, uniform pore size and pore distribution, no evaporation, no effect due to hydrophobic barriers, laminar flow, and a single-phase fluid [50]. In Table 1, are summarized some of the most notorious L–W modified equations that address the phenomena in more realistic scenarios such as the one proposed by Camplisson et al., which includes evaporation effects, Equation (2) [51], Jahanshahi et al. for gravitational effects, Equation (3), [52] the Hong and Kim equation, Equation (4), which account for differences in contact angle due to hydrophobic barriers [53], and the Feng et al. equation that considers the effects due to viscosity and slippage [54]. Several other approaches are extensively reviewed in the research done by Cai et al. [50].

Furthermore, Table 1 presents Darcy’s law and its modifications to describe a system with n-connected sections of varying geometry, the flow rate through the fluidic circuit modeled using an electrical circuit analogy, Equation (7) [55], the Brickman model for flow through highly porous media, Equation (8) [56], the Richards model for the saturation of the media description, Equation (9) [57], and the Elizalde et al. equation to describe the flow into a porous media with variable cross-sectional sections, Equation (10) [58].

Nevertheless, as mentioned in Section 2.2, several materials are often deposited or embedded into the fibrous mats in order to enhance the overall performance of the device. The complexity can be increased by incorporating more degrees of freedom to the system by material properties.

In this regard, models used to recreate or characterize the flow through porous materials might help describe this phenomenon in such complex scenarios. The directional fluid transport can be modeled by the following three processes: Wettability gradient, Janus wettability and a Matrix–liquid combination. The work done by Chowdhury et al. [59] addresses the problem of droplet transport by the effect of wettability gradient and confinement. The problem was defined as the transport of a droplet in an open surface microfluidic device with two variables, wettability and surface confinement. The complexity of the task commands the implementation of finite element method and computational fluid dynamics with the COMSOL tools, the level-set equation [60], denoted with (ϕ), solved is the following:(11)$$\frac{\partial \phi }{\partial t}+U\cdot \nabla \phi =\gamma \nabla \cdot \left(\varepsilon \nabla \phi +\phi \left(1-\phi \right)\frac{\nabla \phi }{\left|\nabla \phi \right|}\right),$$

The equation describes the flow of two immiscible liquids and has two terms; the left side represents the motion of the interface between air and water, while the right side represents the numerical stabilization. In general, the parameter used are interface thickness (ε), the intensity of reinitialization (γ), density, and viscosity of fluids. With this level-set equation, the relation of densities and viscosities are denoted by $\rho =\rho_{1}+\left(\rho_{2}-\rho_{1}\right)\phi$ and $\mu =\mu_{1}+\left(\mu_{2}-\mu_{1}\right)\;\phi$. Alongside, the Navier-Stokes equation is solved with the condition of incompressibility $\nabla \cdot \vec{u}=0$.
(12)$$\rho \frac{\partial \vec{u}}{\partial t}+\rho \left(\vec{u}\cdot \nabla \right)\vec{u}=\nabla \cdot \left[-pI+\mu \left(\nabla \vec{u}+\nabla {\vec{u}}^{T}\right)\right]+\vec{F}+\rho \vec{g}+\sigma \kappa \delta \vec{n},$$
where the expression $\sigma \kappa \delta \vec{n}$ represents the surface tension forces at the interface, more details on the model and simulations are presented elsewhere [60]. The results from Chowdhury showed a map of droplet transport regimes for different wettability gradients and confinements and gave the outline to design microfluidic devices.

The process presented by Wang et al. [61] explains a method to produce a controllable Janus porous membrane for water harvesting. The method of Janus wettability used for water transport has an anisotropic Laplace pressure given by two complementary equations. First, the hydrophobic region of pressure difference:(13)$$\Delta P_{drop}=-2\gamma \cos\frac{\theta }{r},$$
where γ is the surface tension of water, θ is the water contact angle, r is the curvature radius of the water-air interface. Additionally, the second material with capillary function and aids the process by the hydrophilic property is expressed by:(14)$$\Delta P_{capillaryforce}=2\gamma \cos\frac{\alpha }{r_{o}},$$
where γ is the surface tension of water, α is the water contact angle, and $r_{o}$ is the total curvature radius of the water-air interface.

Finally, Mixed-Matrix Membranes combine the transport principles of polymer and inorganic membranes. The mechanism assumes three processes, adsorption, diffusion, and desorption. The chemical potential $\left(\nabla_{\mu }\right)$ gradient gives permeability and selectivity across the membrane and depends on concentration gradient $\left(\nabla c\right)$, and fugacity $\left(f\right)$ conditions towards the permanent flux $\left(J\right)$ which is defined by:(15)$$J=\bar{D}\bar{S}\left(-\Delta f\right)/l=P\left(-\Delta f\right)/l,\;\;\Delta f=f_{2}-f_{1},$$
where the membrane thickness (l), fugacity difference (Δf), and permeability (P) are determined in Barrer [62]. The diffusion and sorption are based on the flux equation defined by the previous equation, while the following two equations describe the diffusion and sorption:(16)$$\bar{D}=\left(1∕c_{1}\right)\int_{c_{2}\approx 0}^{c_{1}}D\left(c\right)dc,$$
(17)$$\bar{S}=\left(1∕f_{1}\right)\int_{c_{2}\approx 0}^{c_{1}}dc=c_{1}/f_{1}.$$

The resolution of these equations is presented elsewhere [63]. The short review of models to recreate and characterize the fluid flow through porous materials resembles its complexity. Additionally, the emergence of smart materials potentiates the ability to manipulate samples to a larger extent.

## 3. Paper-Based Analytical Devices and Electrospun Fiber-Based Biosensors

Paper-based Analytical Devices (PADs) and electrospun fiber-based biosensors (EFBs) have been classified predominantly based on their transduction. Signal transduction encompasses several mechanisms, including optical, electrochemical, thermometric, piezoelectric, magnetic, and micromechanical [64,65]. However, the literature review points out a trend of implementing mostly electrochemical and optical mechanisms for fiber-based biosensor fabrication. Even though this review will focus on transduction mechanisms, it is worth to mention and briefly discuss that cataloging according to the structure of the sensor and how the analyte sample is driven through the device is also popular, as depicted in Figure 3. (a) Dipsticks, (b) lateral flow assays (LFAs), and (c) microfluidic PADs (μPADs) and EFBs are the main types considered in this context [24,66]. The first one is the most straightforward format because the handling of fluids is performed with no control [67]. Biological samples are placed on paper strips containing reagents that trigger an output signal [24].

In contrast, LFAs, μPADs, and microfluidic EFBs promote flow in the desired direction, commonly horizontally, but the last one may also do it vertically [10,24,68]. A classical LFA strip comprises the sample, conjugated, detection, and absorbent pads [24,69]. The sample pad transports and filtrates the sample towards the conjugate pad. The conjugation pad contains the bio-recognition element (BRE). The detection pad containing the test and control lines captures reagents and develops signals. The absorbent pad provides the driving force for fluid flow and prevents the backflow of the sample. Its rapid response, simplicity of fabrication, long shelf life, portability, easiness of use, and low-cost have made LFAs one of the most widespread biosensors currently available.

LFAs have been effectively employed for detecting numerous biological targets and in all kinds of samples such as water, blood, or environmental sources. Nevertheless, LFAs still have to address several challenges, such as sensitive and quantitative detection of multiple targets in the range of pico-molar concentrations, sample pretreatment for non-fluid samples, precision reduction due to inappropriate sample volume, and complexity of antibody preparation [70,71,72]. Consequently, notable efforts are focused to develop high-sensitive LFAs aiming to surpass the aforementioned issues. The detection methods of these devices can be categorized based on their sensing modalities (optical, thermal, magnetic, and electrochemical) that can operate with various tracers (metal or oxide nanoparticles, magnetic materials, polymers and carbon materials), and amplification or recognition strategies (antibodies, enzymes, aptamers, or click chemistry). Extraordinary and detailed research on LFAs fabrication can be found in the work done by Parolo et al. [70] and advanced strategies for detection in the research of Van-Thuan et al. [73].

μPADs and microfluidic EFBs are biosensors that emerged by patterning microchannels over paper sheets or fiber-based mats to create a hydrophilic zone, surrounded by a hydrophobic barrier, that guides liquid samples to flow from an inlet to an outlet to detect a specific analyte [74,75]. A wide variety of these microfluidic devices have been created, implementing different manufacturing techniques and transduction mechanisms to give rise to 2D and 3D structures with other microfluidic channels in shapes only limited by the inventor’s creativity. The 2D configurations are characterized due to the horizontal flow of the liquid sample, for example, from left to right, in a single layer, while the 3D versions use multiple layers that permit the sample to flow horizontally and in the vertical direction. Such architecture is more appropriate to perform numerous assays without interfering with each other [74,76]. The fiber-based biosensors herein considered, either dipsticks, LFAs, or microfluidics, fall on electrochemical or optical transduction mechanisms and, thus, their basics and some related works will be aborded in the following sections.

### 3.1. Electrochemical Transduction

The electrochemical transduction mechanism involves producing an electrical signal due to the interaction between the sensing surface and the target analyte [77]. The generated signal can be characterized through different methodologies classified as amperometric, impedimetric, and potentiometric techniques based on the nature of the output signal [78,79,80]. The performance of a biosensor is experimentally evaluated based on its sensitivity, selectivity, the limit of detection (LOD), linear and dynamic ranges, and reproducibility of the response [81]. These features can be improved by adding a wide number of materials specifically selected or designed, such as nanoparticles, carbon materials, or polymers. Additionally, different techniques might be used to deposit such materials into the fibrous matrix [38]. 

Therefore, in Table 2, we present the current state of electrochemical PADs, their specific transduction technique, the materials used to functionalize the paper matrix, the target analyte and its source, performance features, and their applications. Moreover, in the search for electrochemical fibrous-based biosensors, several electrochemical EFBs can be found. We reported these devices in Table 3, including also the polymer/solvent used for the fiber fabrication, the functional materials added pre- or post-electrospinning, and the function of the fibers, which can be substrate, coating or precursor. We include an Abbreviation section at the end of the article for a better understanding of the Tables content. 

Amperometry is the method where the gain (reduction) or loss (oxidation) of electrons is triggered by applying a controlled potential to a working electrode with respect to a reference electrode. The resultant current is measured, and its magnitude is directly proportional to the target analyte concentration [124]. Electrical current measurement techniques under this methodology are voltammetry and amperometry. When voltammetry techniques are used, a dynamic potential is applied to trigger a current response in the form of a peak or a plateau [80]. In contrast, amperometric techniques, such as chronoamperometry (CA), apply a constant potential to induce a change of current with respect to time [12].

An example of amperometric characterization is exposed in the work of Ruecha and co-workers [86]. They built an enzyme-based PAD for cholesterol detection. The substrate, filter paper, was modified with nanocomposite materials such as graphene, polyvinylpyrrolidone and polyaniline to become the working electrode. The analytical performance was characterized via cyclic voltammetry (CV). The modified paper conferred the PAD good sensitivity, having a 1 μM limit of detection (LOD), but with an inconvenient selectivity. Common interfering molecules in the detection of cholesterol were negligible, except for the ascorbic acid. They opted to add an anionic surfactant to the working electrode to prevent such interference effect. This PAD for cholesterol detection has good selectivity, a wide linear range, and improved selectivity with real biological samples, but efforts to improve reproducibility (relative standard deviations between 0.05% and 9.37%) need to be made.

Pavinatto et al. also applied an amperometric technique for characterizing their EFB. They applied CA for the correlation of 17α—Ethinylestradiol (EE2) concentration with respect to the current magnitude (Figure 4). This biosensor, based on polyvinylpyrrolidone, chitosan and reduced graphene oxide electrospun nanofibers onto a fluorine doped tin oxide paper coated with laccase, exhibited their synergistic electrochemical properties with a limit of detection of 0.15 pmol L^−1^ [110]. This EFB was tested with real and synthetic urine samples and demonstrated its high rate of recovery. However, important analytical features may be due to the fluorine doped tin oxide substrate, which could compromise the device affordability.

Another interesting work is that of Unal et al. [109]. They developed an amperometric EFB based on organoclay nanofibers, consisting of poly(vinyl)alcohol (PVA) and polyamidoamine (PAMAM) dendrimers intercalated within a montmorillonite clay to detect glucose in soft drinks, Figure 5. The resultant device had a limit of detection of 0.7 µM, with no interference effects detected. However, the operational stability was reduced by about 44% within 51 h. Therefore, it is suggested that it should be worked on to improve the overall performance of the device.

Potentiometric methods are based on measuring the electrical potential between working and reference electrodes, while the electrical current is controlled to be almost null. Potentiometry is claimed to be robust, label-free, and with a wide linear range [90,125]. Akanda et al. presented a novel ultrasensitive interference-free potentiometric LFA for the detection of troponin in serum. They took advantage of a pH-dependent electrochemical-chemical-chemical (ECC) redox cycle involving Ru(NH3)_6_^3+^, enzyme product, and tris(3-carboxyethyl)phosphine. The pH dependence of the ECC allowed them to introduce an advantageous combination of an enzyme label, substrate, and products which coupled to a low applied potential (0.05 V vs. Ag/AgCl) minimized the interference of electroactive species present in serum, such as L-ascorbic acid, acetaminophen, and uric acid. This detection strategy had an LOD of 0.1pM-1 of troponin and fast response of 11min after the serum addition [125].

Guadarrama et al. presented an enzyme-linked electrochemical PAD, where the potentiometric method is applied to monitor glucose levels in beverages for the food industry. In Figure 6, the potentiometric response and the calibration curve of this work are shown. The biosensor measurements are performed through a platinum paper electrode functionalized with Nafion that, coupled to a double junction reference electrode (Ag/AgCl/KCl), can minimize interference of typical redox-active anions. This approach enhanced sensitivity to detect glucose with an LOD of 0.02 mM [90]. However, to fall within the linear range of the device, 0.03 to 1 mM, samples had to be diluted, which could represent a complication for the end-user. This PAD is a promising device tested with real samples and giving results within 20–30 s, considerably less time-consuming than conventional methods for this purposes that require up to 30 min [90].

The impedimetric method monitors the impedance change caused by the binding of analytes to BREs or by the metabolites produced during a chemical reaction [126]. Electrochemical impedance spectroscopy (EIS), for example, is a powerful and nondestructive technique capable of characterizing the dynamics of biological interfaces electrically [127]. It accurately describes an electrochemical response of a cell to a sinusoidal voltage signal as a frequency function. The resulting sinusoidal electric current wave I(t) is phase-shifted with respect to the applied potential wave V(t). The ratio V(t)/I(t) is defined as the impedance (Z), which can be interpreted as the resistance to the flow of electrons [128]. The work by Rahimi and co-workers gives an example of an EIS application [102]. They designed a PAD to quantify *L*-tyrosine levels in human blood plasma by coupling CV with EIS. The role of EIS was to optimize the electron transfer property of the electrodes, that is, to minimize their electrochemical impedance for appropriate *L*-Tyrosine sensing [102]. In contrast, Migliorini et al. applied EIS to detect and quantify urea in milk samples within an EFB based on polymeric electrospun nanofibers of polyamide 6 and polypyrrole modified with zinc oxide nanoparticles. The resultant limit of detection was 0.011 mg dL^−1^ [119].

Electrochemical biosensors have inherent advantages such as robustness, ease of miniaturization, selectivity and low detection limits [81,100]. They operate with simple instrumentation, are inexpensive to produce and commonly require small samples [91]. Moreover, they have the potential to be insensitive to lighting conditions and insoluble particles. Thus, they are suitable to be used in both optimal and challenging environments [129]. A major associated disadvantage is the requirement of auxiliary equipment, which may compromise their versatility, transportability, and operation ease [49]. However, these particular issues are continuously being addressed to attend to the appeal of decentralization of testing and treatment of biological samples from specialized facilities. Cánovas et al., for example, developed a device capable of working as a generic potentiometric platform and faced the issue by coupling a wireless miniaturized potentiostat to easily readout the data [107]. The potentiostat allowed the platform to perform quickly in situ measurements and send the assay data to a conventional smartphone, maintaining its simplicity of operation. The platform was proof for glucose monitoring in blood samples and showed a good analytical performance, comparable to the commercially available glucometers but with lower cost and power consumption. However, unlike commercial devices, it requires a calibration procedure and the dilution of the samples.

It is important to mention that great advances are being achieved in developing electrochemical LFAs. The integration of electrochemical readout devices through ink-jet printing, photolithography and screen-printing technologies are turning the LFAs from qualitative and semiquantitative results into reproducible and ultrasensitive fully quantitative biosensors. Like μPADs, electrochemical LFAs use electrodes and redox indicators to recognize charged species for the detection of target biomarkers. However, the integration of electrodes and reagents required for electrochemical reactions increases the cost and complexity of typical LFAs, limiting their immediate application [130].

### 3.2. Optical Transduction

Optical detection is possible due to the interaction of optical fields with BREs [131]. Optical biosensors are classified as label-free and label-based approaches. In a label-free approach, the detected signal is generated directly by the interaction of the analyte with the transducer. In contrast, label-based sensing involves using a label, and the optical signal is then generated by a colorimetric, fluorescent, or luminescent method [131,132]. In this review, colorimetry and spectroscopy, based on phenomena like surface plasmon resonance (SPR), localized surface plasmon resonance (LSPR), surface-enhanced- Raman scattering (SERS), fluorescence, and chemiluminescence, were found in PADs and EFBs. We present some relevant PADs and EFBs with optical transduction in Table 4 and Table 5, respectively, where the readers may find useful information about the devices such as the target analyte, materials, manufacturing process, analytical performance, and applications. We include an Abbreviation section at the end of the article to understand the Tables content better.

Colorimetric detection in PADs and EFBs has received considerable attention due to their operational and readout simplicity [43,46,152]. They are simple, stable, fast, reproducible, easy to be scaled for industrial production, and do not require high technology instrumentation [13,151]. However, some factors may limit its potential such as paper or fiber mat features itself—for instance, the thickness. When the thickness of the paper or mat is large, it may hinder the biosensing task by introducing dull signals, that is, poor visibility due to a large bed-volume [24]. Therefore, the paper or mat thickness must be considered the primary parameter to achieve an equilibrium in mechanical properties and colorimetric response.

The Mahato and co-workers PAD for alkaline phosphatase (ALP) detection is an example of the application of colorimetry as a transduction mechanism. This sensor aimed to provide an instrument-free tool for a rapid preliminary determination of milk quality in the food industry. The immuno-complexation between the sensor-probe and ALP generates blue-green precipitate as an analytical signal (Figure 7). The main advantage of this sensor is that the preliminary readout can be done employing naked-eye inspection. Moreover, semi-quantitative results can be obtained by digital image analysis using a smartphone. This mechanism showed an LOD of 0.87 (±0.07) U/mL and a linear range from 10 to 1000 U/mL [135]. A major disadvantage may be found in the analytical performance dependence on the quality of the image provided by the smartphone camera.

The EFB of El-Naggar and coworkers is another interesting colorimetry-based device [151]. It is a metallochromic sensor for the detection and quantification of ferric ions in water. They used anthocyanin as a spectroscopic probe immobilized into polyvinyl alcohol electrospun fibers. The detection mechanism is based on the formation of anthocyanin-ferric ion coordination complexes that generates a robust and stable colored response within 3–5 s. The color change from colorless to a pink scale is dependent on the concentration of ferric ions. This platform offers qualitative naked-eyed readout and can be coupled to spectrophotometry analysis to quantify the target analytes. However, the quantitative results imply using secondary equipment that may compromise the use of the sensor for point of care testing. Overall, the EFB has a miniaturized interspaced three-dimensional nanofibrous mat with high porosity, good interconnectivity and large surface area that confers an LOD of 7.9 mM and linear range from 17.9 mM to 6.3 mM, that is, good sensitivity, but some selectivity issues to solve.

Among the spectroscopic techniques, SPR and SERS are the most commonly found in biosensing. The SPR phenomenon occurs on the metal-dielectric interface when it is irradiated by polarized light at a specific angle [65]. Such excitation generates oscillations in the charge density known as surface plasmon polaritons and reduces the intensity of reflected light at a specific angle known as the resonance angle. The resultant evanescent field shows extreme sensitivity towards deviations in the refractive index of the environment. During resonance, the incident light beam is absorbed in a particular wavelength or incident angle, resulting in an excitation peak in the signal measured [65]. This effect is proportionate to the mass on the surface. This transduction mechanism is applied in the immunochromatographic PAD developed by Weng et al. for bovine haptoglobin detection [141]. AuNPs, the most common colorimetric label, were used to enhance its sensitivity [10]. When AuNPs are attached to different reagents, such as antibodies, they produce an intense red color due to their surface plasmonic resonance. The detection was triggered by the competitive immunoreaction between the haptoglobin protein and the anti-haptoglobin antibody on the nanocomposite probe. When the target analyte is present in the sample, soft color is appreciated, while in the absence, an intense color appears in the test zone due to nanocomposite accumulation, Figure 8. An LOD of 28 µg/mL was obtained after the calibration, which confers the power for highly efficient rapid on-farm diagnosis [141].

Raman scattering is an extremely weak phenomenon that suggests the inelastic scattering of photons due to their interaction with matter. SERS enhances the intensity of such scattering spectra of a molecule by several orders of magnitude when close enough to nano-roughened metallic surfaces or metallic nanoparticles. For example, Byung Jun and Won-Gun Koh developed an EFB to perform an immunoassay to detect prostate-specific antigen (PSA) using silver nanoparticles to decorate electrospun fibers as capture substrate [146]. Also, AuNPs immobilized with 4-mercaptobenzoic were prepared and used as the SERS tag resulting in a metallic sandwich structure that amplifies the signal by generating hot spots between the AgNPs covered fibers and the gold-based tag (Figure 9). The excellent analytical performance was confirmed by the low and reproducible LOD of 1 pg/mL [146]. Undoubtedly, this platform has good sensitivity and selectivity, but some aspects such as affordability, secondary equipment dependence, ease of operation might limit its application for decentralized applications.

Luminescence is a common optical technique applied to paper-based biosensors. It describes the emission of light as a result of molecules going from their excited to ground state [12]. According to the source providing the energy for molecules to reach their excited state, different luminescent signals can be obtained [153]. For instance, electromagnetic radiance, chemical reactions, heat, frictional forces, and crystallization give rise to fluorescence, chemiluminescence, pyroluminescence, and crystalloluminescence, respectively [153]. The most conventional techniques found in PADs are fluorescence, chemiluminescence, and electrochemiluminescence. 

Fluorescence-based biosensing has gained increasing attention due to its superior sensitivity and its ability to provide simultaneous readouts, for example, intensity, anisotropy, and spectral characteristics [154]. In fluorescence-based detection, analyte molecules or BREs are marked with fluorescent labels. The fluorescence intensity is correlated with the presence of the target analyte and analyte-BRE conjugate interaction strength. Quantitative analysis may result in defiance due to the associated complex labeling processes where the number of fluorophores is challenging to control [9].

Fluorescence can be used as a detection method in PADs. However, there is a major issue to deal with: the inherent fluorescent background of the paper itself [140]. Zhao et al. presented a laser-induced immune PAD for alpha-fetoprotein detection (AFP), Figure 10. Herein, a home-made laser device was used to provide a stable and low-power light source to lower the fluorescent background of paper with good repeatability. Moreover, they enhance the fluorescent signal employing quantum dots (QDs). Since QDs are known to be cytotoxic, they coupled them to silica nanoparticles to overcome this harmful effect. This biosensor achieved good analytical performance for AFP detection with an LOD 6$\;\times \;10^{-15}$ M [140].

Guo et al. developed another whole-cell fluorescence-based biosensor, where *Cupriavidus metallidurans* bacteria was used to obtain a red fluorescent signal in the presence of gold ions from human urine samples on a paper device. The strong colorimetric response allowed the authors to propose a smartphone-based fluorescence diagnostic system with an LOD of 110 nM, indicating a high sensitivity detection [33]. Even when this device presents a novel and clever use of nanotechnology for gold detection, there is no sufficient evidence relating the amount of gold present in tissues or biofluids and its relation to toxicity; thus, the device application might be limited [155].

Chemiluminescence (CL) is attractive for biosensing purposes due to its related simplicity, high sensitivity, good selectivity, low power demands, and cost-effectiveness [156]. Combining this technique with the use of paper and fiber mats is valuable because it avoids the requirement of external energy sources for fluid transport, as it occurs via capillary forces and requires a small volume sample [45].

CL fibrous-based devices have been poorly exploited by research groups and thus have very few literature reports. Herein, a CL-based PAD developed by Li et al. to detect Prostate-specific antigen (PSA) is presented. This device used antibodies as recognition elements coupled to Multi-Walled Carbon Nano-Tubes (MWCNTs) decorated with TiO_2_ NPs to enhance the CL emission. This approach provided an excellent linear response range from 0.001 to 20 ng/mL with an LOD of 0.8 pg/mL under optimal conditions [45].

Electrochemiluminescence (ECL) involves the conversion of electrical energy into radiative energy [157]. It relies on the activation of luminescence by the decay of excited molecules to their ground state, a process where photons are emitted [158]. These molecules are known as luminophores and become excited by applying a voltage to the electrode surface that contains them to trigger an electron transfer reaction [157]. Generally, photon detection is done with photomultiplier tubes (PMT) [159].

ECL provides outstanding advantages like high sensitivity, controllability, reproducibility, and low background [160,161,162]. Nevertheless, multiple manual operation dependence and large-scale and costly instrumentation hamper the ECL development in biosensors [44,163]. For example, Li et al. developed a 3D origami immune-device to detect Carcinoma Antigen 125 (CA 125) with high sensitivity, based on in situ enzymatic reactions to generate H_2_O_2_ as a co-reactant of peroxydisulfate solution with bimetallic palladio-gold nanoparticles (Pd-Au NPs) as a catalyst for signal amplification. The proposed biosensor exhibited high sensitivity and specificity with an LOD of 0.06 mU/mL [164]. Huang et al. developed an autocleaning ECL PAD biosensor for the monitoring of Ni^2+^ and Hg^2+^. Binary catalysis consisting of the intermolecular co-reaction of H_2_O_2_ and N-(4-Aminobutyl)-N-ethylisoluminol (ABEI), and intramolecular catalysis of polyethyleneimine (PEI)-ABEI was applied to reach ultrasensitive detection. The biosensor analytical performance demonstrated high sensitivity, wide linear ranges, and low detection limits (3.1 nM for Ni^2 +^ and 3.8 pM for Hg^2+^) [163]. Yang et al. fabricated a pen-on-paper ECL μPAD where a constant potential triggered a sandwich-type immunoreaction to detect CA199 antigen. Ru(bpy)_3_^2+^–AuNPs (where bpy =2,2′-bipyridine) were used as ECL luminophore. The device contained a hydrophilic paper channel and two electrodes that were fabricated with a 6B black pencil. This approach achieves high sensitivity and stability, corroborated with an excellent linear response range from 0.01–200 U/mL with an LOD of 0.0055 U/mL [44].

Despite the recent advances in paper-based ECL biosensors, some issues need to be addressed to achieve industrial production. For example, new low-toxic eco-friendly materials to enhance sensitivity and shelf-life. Also, more research is required to detect and design new co-reactants to develop ultra-sensitive assays by employing amplification techniques, such as reagents that emit different wavelengths than those from Ru(bpy)_3_^2^ [165].

In general, optical sensors offer valuable features such as simplicity, insensitiveness to electromagnetic interferences, multiplexed detection, low sample requirements, and low signal-to-noise ratio [9]. Apart from these, they show high specificity, sensitivity, small size, versatility, and cost-effectiveness [131]. Many researchers have opted to develop optical biosensors to respond to the growing demand for POC devices in this scenario. Moreover, the continuous evolution of electronics and process power of mobile devices has naturally attracted the attention of researchers in the quest for developing devices able to provide quantitative or semiquantitative results using accessible and non-specialized technology. Figure 11 shows some examples of how mobile technology can be adapted to bioassays to acquire quantitative results.

Figure 11a presents a bioassay integration to quantify creatinine in plasma into a new self-powered pumping system called Self-powered Imbibing Microfluidic Pump by Liquid Encapsulation (SIMPLE) for POC applications. The authors could demonstrate the accuracy and robustness of the sensing platform with an excellent agreement in the clinical range from 0.76–20 mg/dL, low-volume of sample requirements (5 μL), and quick results (5 min) [166]. Figure 11b shows an application developed on Android to measure nitrite concentration a determine pH with a novel combination of a low-cost paper-based device and a smartphone. The pH resolution obtained was 0.04 units of pH, 0.09 of accuracy, and a mean squared error of 0.167. With regard to nitrite, 0.51% at 4.0 mg/L of resolution [167]. Finally, In Figure 11c it is shown a one-step method for the formation of fluorescent silicon nanodots integrated into a reusable, portable, and flexible heater for the determination of total carbohydrates. Using a camera and a smartphone, the researchers could quantitatively analyze the total carbohydrates expressed as an index of glucose or fructose using a greyscale value as the analytical parameter. An LOD of 0.8 μM for glucose and 0.51 μM for fructose was obtained, and a linear response ranging from 10–200 μM [168]. Even though these technological approaches appear to be promising, several issues might be addressed in order to develop a reliable technology. The analytical quantification by these means must consider the light conditions, light source, camera position, type of lenses, and camera parameters such as shutter speed, ISO, and aperture.

## 4. Biosensor Fabrication

Besides the material selection, the fabrication of fibrous-based biosensors requires applying diverse manufacturing techniques to build each component of the device. In this section, we provide a landscape of the trends for fabricating PADs and EFBs from the selection of fiber-based matts and paper to its modification and functionalization to play specific roles, for example, electrodes, reagent storage matrix, microfluidic channels, see Figure 12.

### 4.1. Materials Selection

Since the physical and chemical characteristics of paper directly impact the analytical performance of these analytical devices, paper selection becomes a crucial step. Filter paper is the most frequently used by researchers when fabricating PADs. This selection might be motivated by its ready-to-use state, commercial availability, and excellent capability of absorption. Recently, researchers have focused on fabricating their customized fiber-based paper by electrospinning (ES), a straightforward technique that produces continuous fibers in the micro and nano-scale by applying an electrostatic field to a polymer solution or melts [29,169,170]. In Figure 13a, we present the most common paper sources found in our literature review. Figure 13b shows the nature of the material that composes the different paper sources. Even though a vast number of natural and synthetic polymers were employed for paper fabrication, cellulose continues to be the most popular material present in PADs. 

Besides the material used as a fibrous mat, several materials can be selected to improve the performance of the PADs or EFBs. Advanced functional nanomaterials can be highly selective and sensitive and could be used to construct powerful tools as electrochemical sensors with biomedical applications to detect target molecules as biomarkers of diseases [171].

#### 4.1.1. Filter Paper

Filter paper commonly refers to the "general purpose "qualitative type group. Its applications are regularly related to separation for the determination and identification of materials. Qualitative filter papers are composed of cellulose and available in different grades according to particle retention and flow rate [22]. This research points out that the most used qualitative filter paper grades range from 1–6 in different applications, including PADs fabrication (Figure 13a). Filter paper surface is usually modified or functionalized to improve its physical or chemical properties. Some of the techniques used for this purpose will be further discussed [89,172,173]. 

In Figure 14, there are some examples of filter paper-based PADs. Figure 14a shows the design, components, and principle of detecting an electrochemical PAD to detect double-stranded DNA. AuNPs were used to immobilize methylene blue tagged to triplex-forming oligonucleotides with a capacity of up to eight probes [174]. This device built by Cinti et al. shows an LOD of 3 and 7 nM for single and double-stranded sequences, respectively. Moreover, their homemade approach displayed repeatability with a deviation of around 10%, including tests on undiluted serum samples. However, when working with serum samples, the sensitivity was decreased due to the proteins and fats stacked into the electrode.

Figure 14b shows an electrochemical PAD integrated with nanobioprobes onto graphene film for ultrasensitive multiplexed detection of cancer biomarkers [175]. Figure 14(b1) shows the preparation of nanobioprobes through the co-immobilization of horseradish peroxidase (HRP) and antibody onto monodispersed SiO_2_ NPs, and Figure 14(b2) depicts the schematic representation of the fabrication and assay procedure used to prepare the PAD with bovine serum albumin (BSA) and alpha-fetoprotein (AFP). This device identified four cancer biomarkers in serum samples from cancer patients with good stability, reproducibility, and accuracy. Moreover, the μPAD can be reused up to 24 times by dissociating the antigen-antibody with 0.1 M glycine-HCl.

#### 4.1.2. Electrospun Mats

ES has arisen as a powerful technique that combines versatility, efficiency, straightforwardness, and affordability to produce complex micro and nanofibrous matts of many materials, including polymers, composites, and ceramics [176]. Figure 15 presents a schematic representation of the electrospinning process. There are basically three components to perform the process: a high voltage power supplier, a needle, and a collector [177]. When the polymer solution or melt is subjected to a critical electrical force, this one overcomes the surface tension, and a polymer jet is ejected towards the collector. On its way from the tip of the needle to the collector, the solvent evaporates, producing non-woven nanofibers with high specific surface area [28]. 

Moreover, the addition of carbon-based materials, metal and ceramic nanoparticles and functional polymers can produce fibers suitable to attach selectively sensing biomolecules, enhance conductivity in EC sensors, or improve colorimetric response [7]. In this regard, a wide number of publications can be found. In Figure 16a, it is presented a modification to the typical ELISA (Enzyme-Linked Immunosorbent Assay) immunoassay by placing polyhydroxy butyrate (PHB) electrospun membranes coated with polymethyl methacrylate-co-methacrylic acid, poly (MMA-co-MAA) in the bottom of the microtiter plate. With this approach, Hosseini et al. were capable of detecting dengue virus with higher sensitivity than conventional ELISA immunoassay. This superior sensitivity was achievable due to the advantageous combination of large specific surface area available for biomolecular interaction due to PHB nanofibers and the presence of surface carboxyl (–COOH) groups from MAA segments for covalent and physical protein immobilization. Even though the sensitivity of the device was outstanding (up to 100%), the device lack of accuracy and specificity, having 84.6 and 80%, respectively, with their best configurations [136].

Moreover, the resourcefulness of ES allows the production of ceramic and metallic nanofibrous composites. Yun et al. developed a metal-enhanced fluorescence biosensor, Figure 16b. The electrospun poly(caprolactone) (PCL) decorated with photo-reduced silver nanoparticles coated with silica. The function of the silica layer was to create controlled spaces between the AgNPs and the fluorescent molecules to tailor the fluorescent response. Their results were compared with similar publications obtaining at least thirty times the LOD (1.5 ng mL^−1^) of BSA. However, the authors believe that the metal-enhanced fluorescence can be further improved by controlling the type, size and shape of the metal-based nanostructures deposited on the fibers [142].

Finally, another great advantage of ES fibers is that the polymer blend can be tailored a priori and thus obtain fibers with desired characteristics. In its work, Shepherd et al. controlled the amount of poly(lactic acid)/poly(lactic acid)-*b*-poly-(ethylene glycol) (PLA/PLA-*b*-PEG) and PLA/PLA-*b*-PEG-Biotin fibers in water. The used solvents influenced the stability of biotin in water. Biotin is widely used in biosensing applications that require rapid protein binding. The obtained fibers improved the stability of biotin in water, and fibers formed using 1,1,1,3,3,3-hexafluoro-2-propanol (HFIP) have greater stability than fibers spun from DMF. From the initial 7.6% of initial biotin, they reduced the biotin migration to the aqueous phase retaining 2% after seven days of water exposure [178].

#### 4.1.3. Novel Materials

A wide range of novel materials has been reported on biosensors fabrication, such as metal nanoparticles, metal oxide nanomaterials, carbon nanomaterials, polymers matrixes, metal-organic frameworks, nanocomposites, among others. The advances in the materials science field might trigger a new generation of biosensors soon, as is analyzed in this section.

Metal nanoparticles based on gold, silver, platinum, and palladium and their corresponding alloys offer superior properties to design biosensors. Gold nanoparticles (AuNPs) are widely useful for fabricating biosensors due to their good biocompatibility, excellent chemical stability, and convenient surface modification; they are frequently used as indicators in colorimetric assays [38]. These advantageous features of AuNPs allowed researchers to develop biosensors for a wide number of target analytes such as trichloropyridino [179], olaquindox residues [180], DNA [181], thrombin [182], carbofuran [183], glucose [179], and cancer cells [184]. 

Metal oxide nanomaterials, such as cerium oxide, copper oxide, nickel oxide, titanium oxide, or zinc oxide, are employed due to the huge surface area and its high sensitivity and selectivity. Zinc oxide nanoparticles possess unique properties such as high catalytic efficiency, biocompatibility, and non-toxicity with high isoelectric points [185]. Among the metal oxide nanostructures, iron oxide has promising applications for cancer biomarkers detection due to its fast electron-transfer kinetics, strong adsorption, high surface-to-volume ratio, non-toxic nature, and biocompatibility [75].

Carbon nanomaterials include graphene, fullerenes, multi-walled carbon nanotubes, single-walled carbon nanotubes that offer electrical conductivity, chemical durability, and biocompatibility. Graphene nanosheets and carbon nanotubes enhanced electrical conductivity. They can also immobilize biomolecules, improving the active surface area and plasmonic fields around AuNPs and AgNPs, resulting in fluorescence quenching and color changes [186]. Functionalization on paper-based biosensors surfaces incorporating nanomaterials and nanostructures to enhance their performance, adding sensitivity, dynamic range, limit detection, and selectivity.

Polymer nanomaterials based on electrically conductive polymers, dendrimers, and molecular imprinted polymers are used to detect biomolecules due to their molecular shape recognition and biofunctionalization, increasing electrochemical signal transduction. Ionic liquids and conductive polymers have been displayed as potential smart and low-cost materials for biosensors development. Ionic liquids are composed of bulky organic anions and cations that provide the chemical sensitivity and selectivity for detection based on the differential capacitance. Conductive polymers are materials that can be tuned, altering conductivity and solubility, thus improving wettability in ionic liquids. Smart composite materials combining the properties of both ionic liquids and conductive polymers components can enhance the response time, selectivity, stability and sensitivity [187].

Metal-organic frameworks (MOFs) are crystalline nanoporous coordinated materials composed of metal ions/secondary building units (SBUs) with organic linkers [188,189,190]. The MOFs by themselves can be directly used in biosensing applications for colorimetric or electrochemical detection. Moreover, MOFs combined with other materials have introduced novel structures with desirable features for biosensing applications, such as excellent chemical stability, sensitivity, flexibility, and specificity [191]. MOFs can be used to carry sensitive elements, they can emulate the function of enzymes, and can improve the electrochemical and optical signals [192]. 

Nanocomposite materials developed with biopolymers such as proteins and nucleic acids and metals or carbon nanomaterials displayed properties for biosensors applications because of their electronic and catalytic behavior, selective recognition to detect biomolecules, drug delivery, and bio-imaging. Micro-RNA is a biomarker of cancer expression, early detection of microRNA is widely important in the diagnosis, treatment, and prognosis of the disease. An *N*-carboxymethyl chitosan/molybdenum carbide nanocomposite as electrode material was reported for micro-RNA detection with selectivity and reproducibility [193].

### 4.2. Paper Functionalization Techniques

Frequently, the material components of PADs comprise cellulose, nitrocellulose, and hemicellulose. These materials are current insulators and thus inhibit electrical signals in EC PADs [24]. Moreover, even though cellulose brings many hydroxyl functional groups, they are not reactive enough towards biological elements. Even electrospun mats require a surface modification if the functional materials were not added into the polymer solution before electrospinning [194]. As a result, in most PADs, chemical or physical surface modification of the paper is required to overcome these drawbacks [22]. Some techniques have successfully addressed this challenge. In this work, we focused on techniques frequently reported in recent research publications, such as drop-casting (DCA), screen printing (SP), dip coating (DC). However, other techniques like sputtering, photolithography, inkjet printing, direct writing, flexography printing, and others are also found in the literature. It is worth mentioning that, even though EFBs are most commonly functionalized by the selection of polymers and materials prior electrospinning process, the polymer mats can be submitted to techniques such as DCA, SP and DC, as reported in Table 3 (column: Functionalization post-ES). Nevertheless, we found that wax printing and inkjet printing are methods, applied apparently exclusive to typical paper, at least for biosensing purposes.

#### 4.2.1. Drop Casting

Generally, three steps are involved when a material is coated via DCA: (1) Sample preparation; in this stage, the desired material is mixed in a suitable solvent where sonication prevail as the most recurrent mixing method for non-biological molecules, on the other hand, magnetic stirring is preferred for the mixing of biological elements; (2) Sample cast by dropping on the target substrate; and (3) thin film formation by evaporation of the solvent [85,86,195,196]. The simplicity and ease of preparation of DCA highly favored it over techniques like abrasive immobilization or electrochemical deposition for prior preparations of electrodes in nanomaterial electrochemistry [196]. However, DCA present various disadvantages; for example, it present limitations for covering large areas, the thickness of the film is hard to control and usually present poor uniformity [197]. These drawbacks can be, in some way, controlled or reduced by the regulation of concentration of the solute, combination of solvents, or by heating the substrate to speed up the evaporation process [198,199]. 

Since the DCA principle is driven by surface wetting, not only the deposition of functionalizing elements can be done. In its work, Arduini et al. used this technique to prepare electrodes doped with carbon black and PBNPs for electrochemical detection of several pesticides [47]. However, Stefano Cinti and collaborators went further by synthesizing the PBNPs in situ, adding the required precursors and using the paper as a scaffold for chemical reactions [89]. Finally, it is worth pointing out that DCA can be executed combined with other functionalization and microchannel fabrication techniques like WP, DC, Folding, or SP, as required [47,82,85,86,105,165].

#### 4.2.2. Screen Printing

Screen printing (SP) is a high-performance technique widely used for the fabrication or modification of films [200]. The SP process involves the deposition of the selected paste-like ink through a screen mesh and then through a predesigned photo-emulsion mask, under a determined squeegee pressure Figure 17. The mask blocks the non-image areas on the screen, thus allowing to print selected substrate areas [201]. It is attractive for being low-cost, environmentally friendly, fast, and versatile. The versatility is attributed to the possibility of printing into different substrates, including paper, using a wide variety of materials in the form of inks. However, it presents important limitations, for example, a resolution limit of 100 µm [201], the use of heavy solvents to avoid mesh cells blocking during printing [202] and the considerations of rheological properties of materials intended to be used as inks to maintain the resolution [203].

Some authors have reported using these techniques to deposit conductive inks to manufacture electrodes with biosensing applications. The most commonly used are made of carbon [88,104], silver chloride [100,104] and indium tin oxide (ITO) [204].

Cincotto et al. developed a μPAD for simultaneous detection of uric acid and creatine by electrochemical means. They reported the SP fabrication of the three-electrode system using Electrodag 423SS carbon ink for working and counter electrodes and Ag/AgCl ink for the reference electrode [106]. Cinti et al. fabricated a biosensor to detect nerve agents electrochemically. They claimed to manually fabricate their electrode system with SP using Ag/AgCl Electrodag 477 SS ink for the pseudo-reference electrode and Electrodag 421 carbon ink containing CB/PBNPs powder for working and counter electrodes [103]. Rahimi-Mohseni et al. also reported the use of SP to fabricate the working, counter, and reference electrodes for electrochemical detection of L-tyrosine using graphite, carbon ring, and silver chloride, respectively [102].

Furthermore, the use of hydrophobic agents as ink for the fabrication of microchannels has been reported, for example, wax [205]. Juang et al. reported the fabrication of a μPAD for glucose detection using polymethylmethacrylate (PMMA) to construct the microchannel walls. They proposed the assistance of SP utilizing vacuum filtration and obtained favorable results in terms of time and reliability [206]. Liu et al. fabricated a chemiluminescence biosensor for Listeria monocytogenes detection using wax SC to fabricate de microchannels [207].

#### 4.2.3. Dip Coating

Dip coating (DC) is a straightforward and inexpensive technique extensively used in a large number of industrial fields to deposit a film onto any substrate, including ceramic, metallic, polymer films, and fibrous materials. The process comprises the deposition of a liquid phase containing coating a target material all over the surface of a substrate by the direct immersion of the substrate into the liquid, following the evaporation of the wet sedimentary coating to obtain a dry film [208]. Although the formation of the film is seemingly simple, complex physical and chemical variables are involved. The thickness and morphology of the deposited thin films are governed by parameters such as immersion time, dip coating cycles, withdrawal speed, density, surface tension, viscosity, substrate surface, conditions of evaporation, among others [209].

Kumar et al. analyzed paper modification with reduced graphene oxide (RGO) for cancer detection in their work. Their results comprise the use of poly (3,4-ethylene dioxythiophene): poly(styrene sulfonate) (PEDOT:PSS), RGO composites, and a vast number of solvents like methanol, ethylene glycol, and H_2_SO_4_ to improve the electrical conductivity of the modified paper. In all cases, the devices were prepared using DC [210]. For a completely different purpose, Deng et al. described a protocol based on adsorption and cross-linking of poly(oligoethylene glycol methacrylate) (POEGMA) derivatives to decrease the non-specific protein absorption in PADs. They use a sequential dipping method to deposit the reagents [211]. 

#### 4.2.4. Inkjet Printing

Inkjet printing is a non-contact technique commonly used on PADs to manufacture electrodes, immobilization of biorecognition elements, and even fabrication of microfluidic channels [83,212]. It consists of the deposition of small quantities of ink, in the range of pico-liters, at defined spots on the surface of a substrate employing thermal, piezoelectric or electrohydrodynamic actuators [213]. The variety of inks go beyond dyes and pigments, including hydrophobic and conductive materials. Likewise, substrate possibilities are broad. Commonly, flexible substrates are used, such as PEN, PET, and paper; however, rigid and even 3D substrates are used [212].

Inkjet printing has gained increasing attention due to its flexibility, low cost, and scalability to mass production. Additionally, inkjet printing is promising for both patterned substrates and the deposition of chemical or biological reagents. Nevertheless, its potential is limited by the rheological properties required for the printability of inks [212,214].

Rosati et al. also proved the feasibility of inkjet printing by fabricating a label-free biosensor to detect antibiotics in milk. The technique was used to fabricate a silver NPs based electrode array onto paper and polymer substrates [215]. Likewise, Wu et al. created colorimetric strips to detect acetylcholinesterase inhibitors; the sensing zone was fabricated with an inkjet printing method using a sol-gel derived silica ink, where the biorecognition elements were immobilized [216]. In the work of Ihalainen et al., biotin-functionalized polythiophene films were inkjet printed onto ultrathin gold film electrodes to act as the biorecognition zone for streptavidin protein [217]. Similarly, Abe et al. used this technique to fabricate a microfluidic pattern onto filter paper and immobilize the reagents necessary for the chemical sensing of pH, total protein, and glucose in a simultaneous mode [213].

#### 4.2.5. Wax Deposition

Several techniques can be implemented when fabricating hydrophobic channels using wax, such as screen printing [46,218], dipping [91,219], stamping [220], and hand-drawing [46,221]. However, recently wax printing has become extensively used to fabricate microfluidic devices made of paper for being a fast-prototyping technique. Fundamentally, it requires a printer and a hot plate. The printer deposits solid wax according to the previously assigned design: the hot plate melts the solid wax to penetrate through the paper thickness to create complete hydrophobic barriers [37]. Besides being fast, simple, inexpensive, and with a high throughput production [221], wax printing has many advantages. It does not require the exposure of hydrophilic channels to polymers or solvents [67].

Moreover, it has been noticed that the use of wax printing reduces the electrochemical background signal when compared to photolithography resulting in better PAD performance [222]. However, wax printing possesses some drawbacks. One of them is that when the wax melts, it spreads vertically and horizontally into the paper. The vertical spreading creates the hydrophobic barrier across the thickness of the paper. In contrast, lateral spreading becomes an issue because it decreases the resolution of the printed pattern resulting in microchannels narrower than expected [205]. The minimum feature sizes obtained with wax patterning have been reported to be higher than the products of photolithography, which is generally between 500 and 800 µm [223]. Besides, wax-printed devices are not suitable for handling surfactants or organic solvents: this situation might result in hydrophobic barrier damage and, consequently, in a loss of sample [223]. Even so, it is the most used technique.

### 4.3. Bio Immobilization Approaches

According to the assay type, a biosensor may use recognition elements such as enzymes, antibodies, DNA/RNA probes, whole cells, and even an artificial version [24]. There are various approaches to perform the bio-attachment and can be generally classified as (1) Physical methods, where the biomolecule is confined into the paper surface only by weak forces such as Van der Waals, electrostatic, and hydrogen and hydrophobic interactions; (2) Chemical methods, where the surface is chemically modified in order to create a covalent bond with the biomolecule; and (3) Biological methods where the biomolecule is attached to paper due to biochemical affinity [39,40,41].

Figure 18a comprises the three categories of bio-attachment and their most common subcategories. Besides, in Figure 18b, there is a comparison between production characteristics such as preparation time, preparation complexity and cost, and characteristics of performance like bonding strength, reversibility, risk of bioactivity loss, and orientation controllability of each method. The qualitative comparison was made based on the judgment by the authors of several articles and reviews on the topic [12,26,39,40,42].

Biochemical immobilization is more expensive, time-consuming and complex to perform because of the modification of both biomolecules and substrate. However, these drawbacks are compensated by attractive features, such as orientation controllability that maximizes the target analyte retention and a fully retained biological activity [26]. On the other hand, physical immobilization approaches are straightforward and inexpensive. Nevertheless, this methodology might present a high loss of bioactivity, lack of orientation, and the weakest bonding strength of all approaches; in some cases, up to 40% of immobilized biomolecules can be desorbed [39]. Finally, chemical immobilization possesses characteristics between physical and biochemical methodologies due to the vast number of techniques that can be performed. It is worth mentioning that the strongest bonds between cellulose and biomolecules are obtained by this method, and the attachment is practically irreversible. However, chemical modifications may induce structural changes in biomolecules resulting in a loss of bioactivity and therefore a loss of sensitivity [224].

## 5. Conclusions and Future Perspectives

In this review, a comprehensive analysis on the most recent progress on paper-based analytical devices (PADs) and electrospun fiber-based biosensors (EFBs) gives an insight into making decisions for the design and fabrication of these biosensors aiming to fulfill the criteria proposed by the World Health Organization (WHO) for resource-limited environments. The fibrous nature of these materials and the easiness to be physically and chemically modified by simple and well-established technologies could lead to the production of breakthrough appliances capable of improving living standards in developing countries.

Despite the compelling features this technology offers, some barriers still prevail and limit the recognition and social impact these assays might provide, including the complete integration of the multiple steps involved in biosensing into a single system and the translation from proof-of-concept devices to standardized protocols. Nevertheless, the detailed inspection of the current trends in PADs and EFBs glimpsed a guide for the selection of materials, bio-immobilization approaches, fabrication techniques, mathematical models, and readout methodologies to accomplish the construction of a complete and robust biosensing platform. Moreover, the side-by-side evolution of the material sciences, nanotechnology, and mathematical analysis combined with increasing computational power is projecting the PADs and EFBs as an astounding tool to precisely manipulate and analyze biological entities. Furthermore, the smart engineering applied to design these analytical platforms allow to take advantage of easy mass-scalable production techniques such as screen or wax printing, while more complex, expensive, and time-consuming techniques, such as sputtering or photolithography, are falling into disuse.

Finally, as the research points out, great efforts are dedicated to developing biosensors using both cellulose and synthetic polymers. Some researchers decided to go for the ubiquity, low-cost, and surface customizability of cellulose paper; others, for the tunable properties of polymers, prior or post electrospinning of the fiber mats. After this review, we conclude that both PADS and EFBs can perform as ultrasensitive sensors for a specific application. If not, their fibrous modifiable nature confers researchers the possibility to play with related variables, like materials and geometries, to tune in the properties to meet the criteria and operate with the desired behavior.

Aiming to attest to such capabilities of PADs and EFBs, we bring out some reported sensors for a specific task: the detection and quantification of glucose [88,90,92,93,94,95,96,97,98,99,107,108,111,113,114,120,146,152,225,226,227,228,229,230,231,232,233,234,235,236,237,238,239,240,241,242,243]. Glucose sensing platforms are among the most common and studied devices, and a wide variety is already commercially available. In Figure 19, the limits of detection (LOD) of several glucose PADs and EFB are plotted. Most of them were developed for clinical diagnostics purposes; therefore, the limits of quantification (LOQ) of commercially available glucometer strips are also included. Moreover, the adult normal physiological range of glucose in the blood, 3.5–5.5 mM [225], and the common operational range of commercial glucometers, 3–20 mM [226], are highlighted as a benchmark. It is worth mentioning that even when glucose is the common analyte in all sensors, the source and concentration of the sample, recognition elements, mediators, transduction mechanism, or even the final application might differ drastically from one to another.

Figure 19 suggests that PADs and EFBs are more sensitive than the commercial glucometers herein considered. However, this fact is not enough to ensure better performance than that of the commercial platforms. Other parameters assessing the analytical performance, especially the linear range, are crucial to consider when arbitrating the operation of biosensors for a specific task. Besides having an LOD or LOQ under the lower limit of the normal physiological range, platforms must be qualified to operate linearly, at least in the purple highlighted region. For example, the potentiometric platform developed by Cánovas et al., in which a Nafion membrane is used to control the mixing potential, can detect a small glucose concentration as 0.1 mM but behaves linearly from 0.3 to 3 mM [107]. Despite being sensitive enough, regrettably, its linear range is not suitable for glucose sensing in blood samples [107]. However, in a recent work, Cánovas et al. modified their material selection, going from Nafion to Aquivion, resulting in a platform with slightly greater but proper LOD (0.16 mM) with a functional linear range of 0.5–10 mM that matches the range of interest for clinical glucose determination in blood samples [108]. Another example exhibiting the tunning capabilities of these sensors is reported in the work of Apetrei et al., where they claimed to change the analytical performance of their platform by incorporating montmorillonite to the PANi solutions before the electrospinning of the fiber mat; the key component of their platform. In general, the PADs and EFBs herein have better performance than the commercial ones, including higher sensitivity and a wider effective linear range.

Finally, we can conclude that, from their conception, fibrous-based biosensors have been perfecting and getting closer to address the need for disease diagnostics, screening, and monitoring to provide equitable healthcare to every individual. Moreover, as the researchers keep broadening the materials, recognition elements, mediators, and transduction mechanism combinations to develop outstanding fiber-based biosensing platforms, the reality of becoming the new gold-standard in POC applications seems closer.

## Figures and Tables

**Figure 1 biosensors-11-00128-f001:**
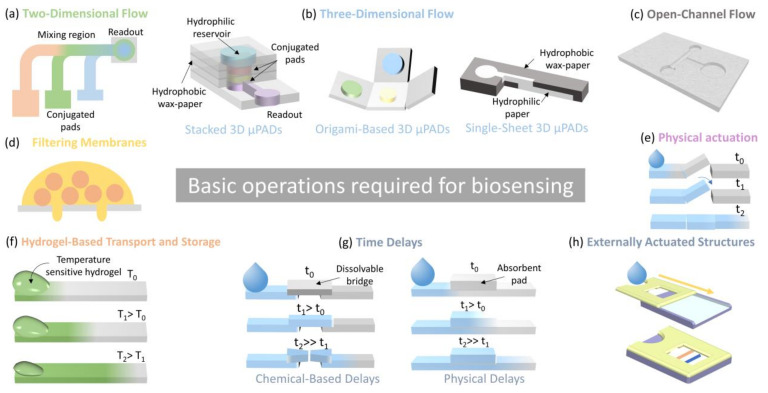
Basic diagnostic operations that fibrous mat can be perform. (**a**) Two-dimensional flow design for sequential delivery of reagents or mixing [14]; (**b**) three-dimensional flow configurations [15,16,17]; (**c**) open-channel microfluidic omni phobic paper, adapted from [18]; (**d**) Separation membrane [19]; (**e**)“on/off” fluidic switch [13]; (**f**) hydrogel-driven paper-based microfluidics [20]; (**g**) flow time delays using dissolvable bridges and absorbent pads [10]; (**h**) slip device for one-step point-of-care testing [21].

**Figure 2 biosensors-11-00128-f002:**
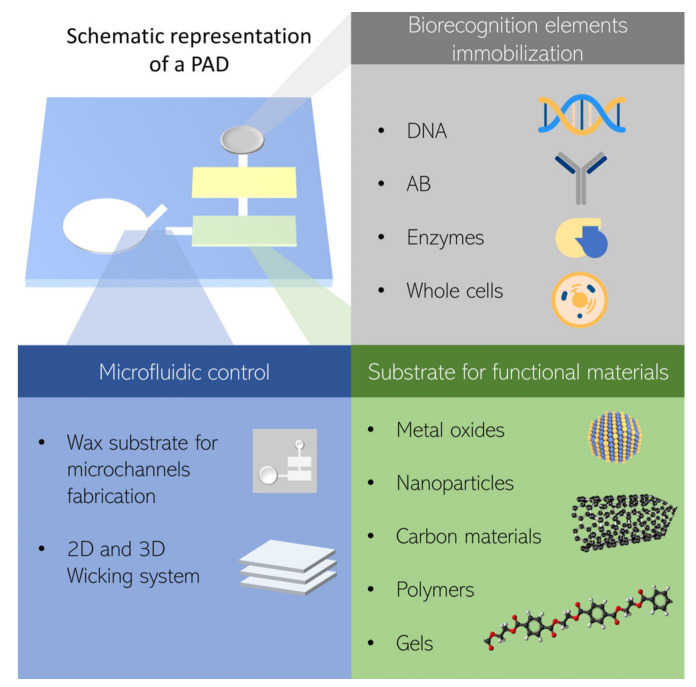
Schematic representation of a paper-based analytical device (PAD) and its primary function as a matrix for biomolecules immobilization, functional materials, and microfluidic control.

**Figure 3 biosensors-11-00128-f003:**
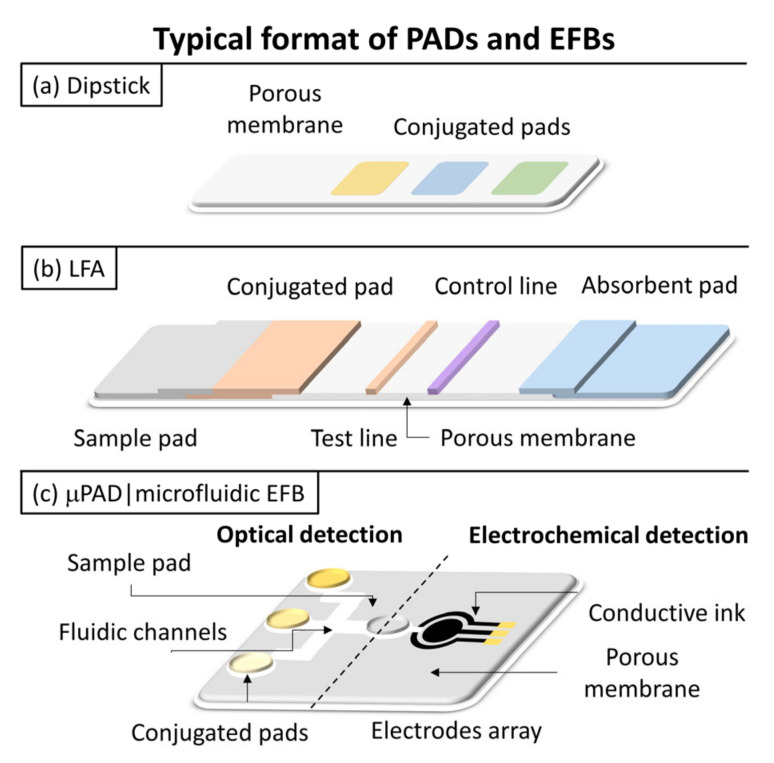
The typical format of paper-based and electrospun-based analytical devices.

**Figure 4 biosensors-11-00128-f004:**
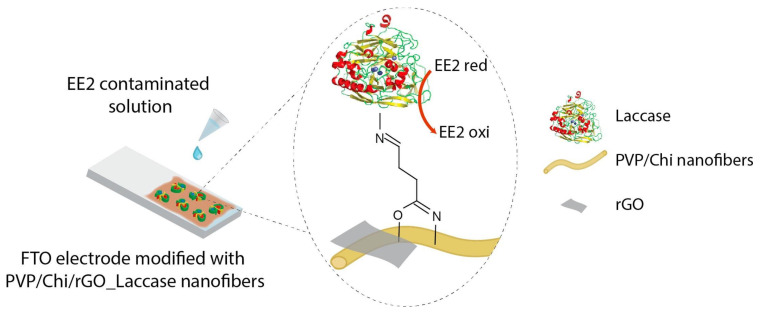
A highly sensitive electrochemical biosensor for EE2 detection, developed by Pavinatto et al. [110]. Reproduced with permission from [110].

**Figure 5 biosensors-11-00128-f005:**
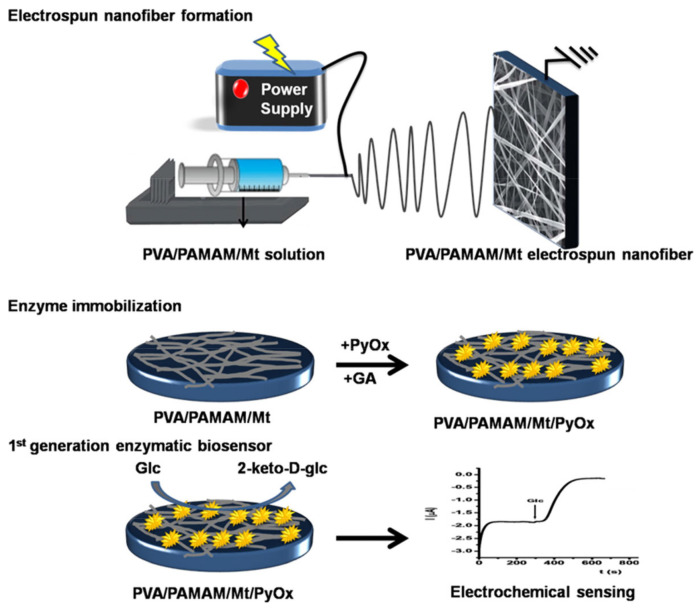
Schematic representation of the preparation of poly(vinyl) alcohol (PVA)/poly(amidoamine)-montmorillonite (PAMAM-Mt)/pyranose oxidase (PyOx) biosensor. Reproduce with permission from [109].

**Figure 6 biosensors-11-00128-f006:**
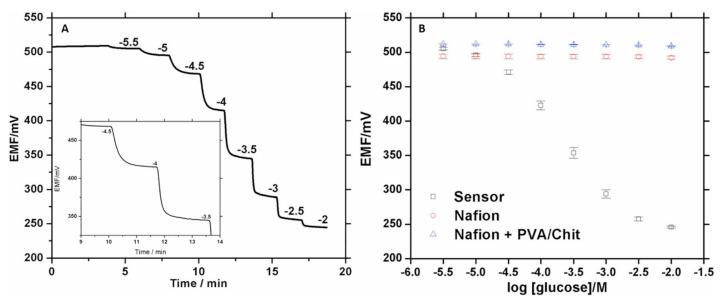
Potentiometric response for the glucose sensor. (**A**) Time trace for the sensor upon increasing glucose concentration and (**B**) calibration plot for the sensor and blank electrodes (mean ± S.D., N = 3). Reproduced with permission from [90].

**Figure 7 biosensors-11-00128-f007:**
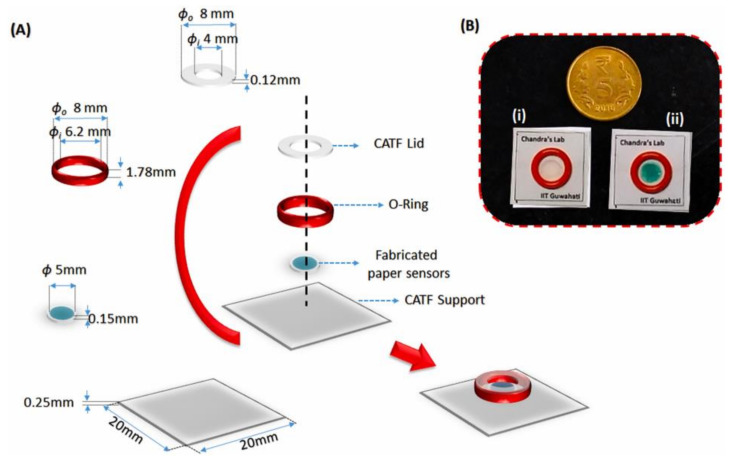
(**A**) A detailed description of a prototype device for alkaline phosphatase detection is based on colorimetry with dimensions and (**B**) the actual prototype showing the color change in the (i) before and (ii) after interaction of alkaline phosphatase (ALP) with the filter paper (P)/4-carboxybenzene diazonium (DS)/3-(Ethylimino methyleneamino)-N,N-dimethylpropan-1-amine-N-hydroxy succinimide (EDC-NHS)/Anti-ALP sensor-probe. Reproduced with permission from [135].

**Figure 8 biosensors-11-00128-f008:**
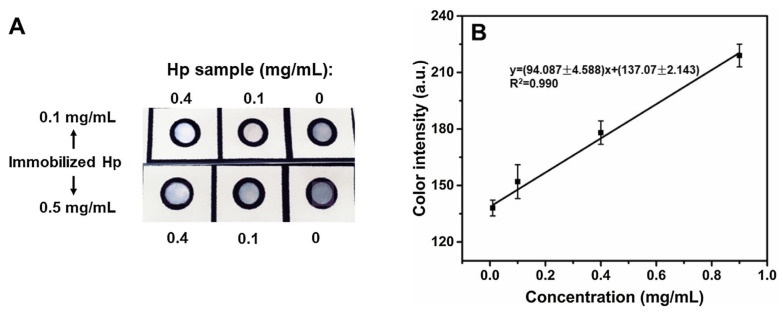
(**A**) Representative resulting test zone images of the paper-based biosensor. (**B**) Calibration curve of Hp protein derived from presented vertical flow immunochromatographic biosensor. Error bars: standard deviation (n = 3). Reproduced with permission from [141].

**Figure 9 biosensors-11-00128-f009:**
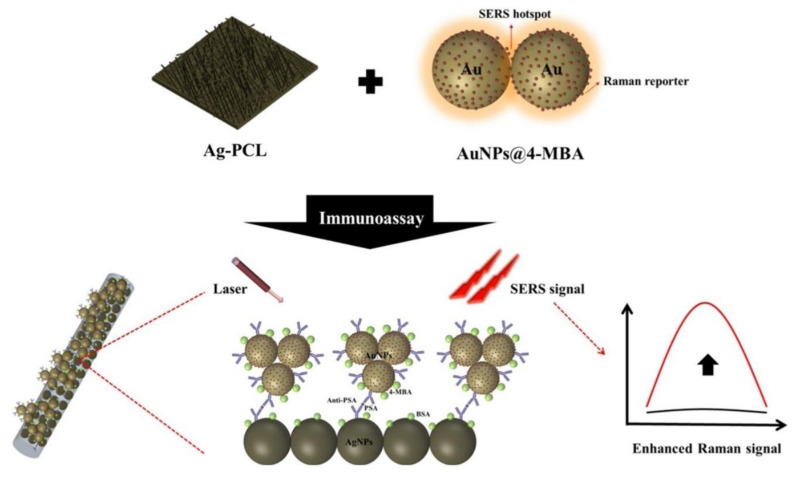
Surface-enhanced- Raman scattering (SERS)-based immunoassay to detect prostate-specific antigen (PSA). Reproduced with permission from [146].

**Figure 10 biosensors-11-00128-f010:**
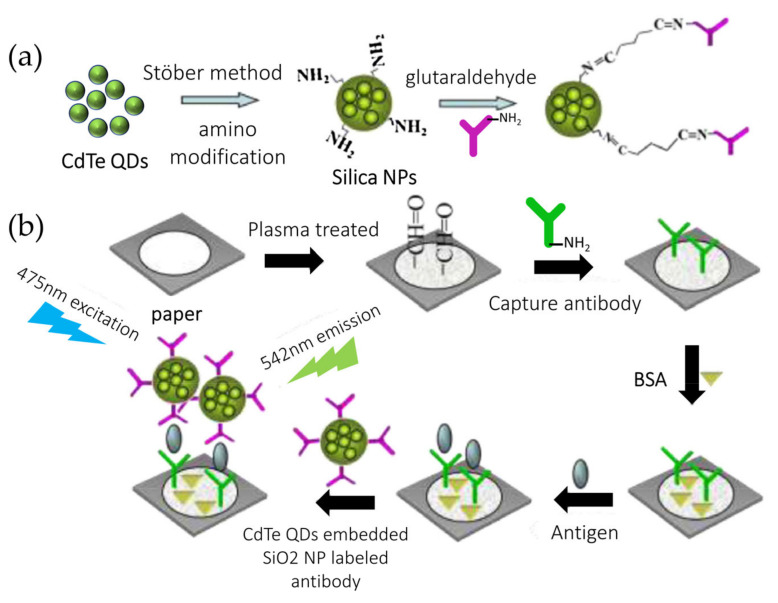
Paper-based laser-induced fluorescence immunodevice. (**a**) Preparation of cadmium telluride quantum dots (CdTe QDs) embedded SiO_2_ nanoparticle labeled antibody. (**b**) Schematic representation of the immunoassay and LIF detection procedure on the paper-based chip. Reproduced with permission from [140].

**Figure 11 biosensors-11-00128-f011:**
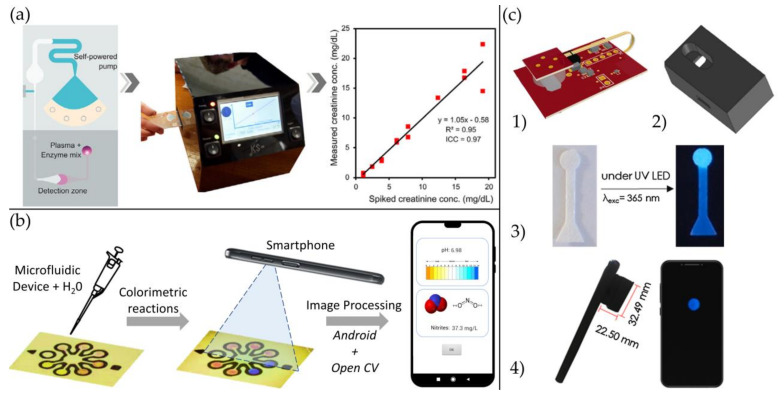
(**a**) Creatinine detection in plasma using SIMPLE technology. Reproduced with permission from [166]. (**b**) Smartphone-Based Simultaneous pH and Nitrite Colorimetric Determination for Paper Microfluidic Devices. Reproduced with permission from [167]. (**c**) Paper microfluidic device combined with laser prepared graphene heater for total carbohydrates determination using a smartphone for quantitative analysis. (**c1**) Device for temperature control (board) inside the housing; (**c2**) Housing made for the board; (**c3**) Microfluidic paper-based analytical device for determination total carbohydrates with synthesized silicon nanodots; (**c4**) Portable housing accomplished to the smartphone. Reproduced with permission from [168].

**Figure 12 biosensors-11-00128-f012:**
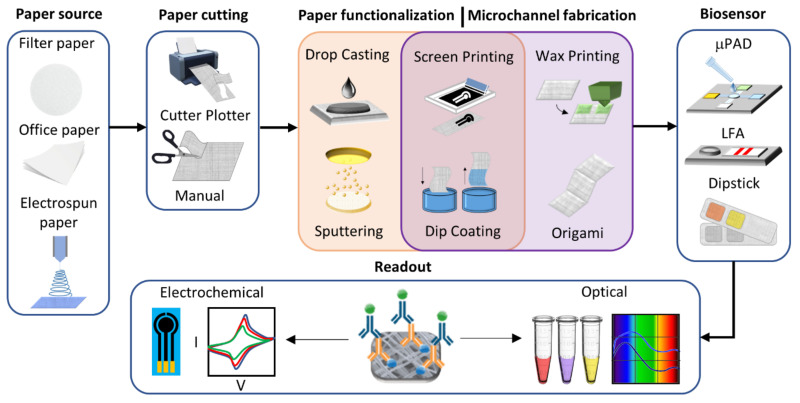
Schematic representation of a PAD or EFB fabrication.

**Figure 13 biosensors-11-00128-f013:**
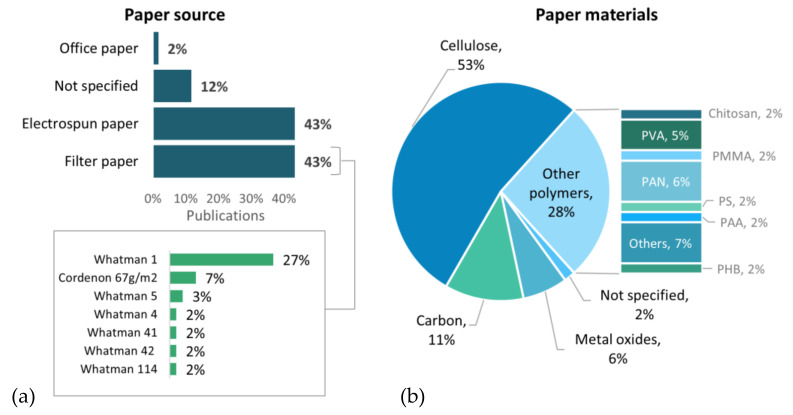
(**a**) Paper source and (**b**) paper materials used for PADs and EFBs fabrication according to the presented literature review.

**Figure 14 biosensors-11-00128-f014:**
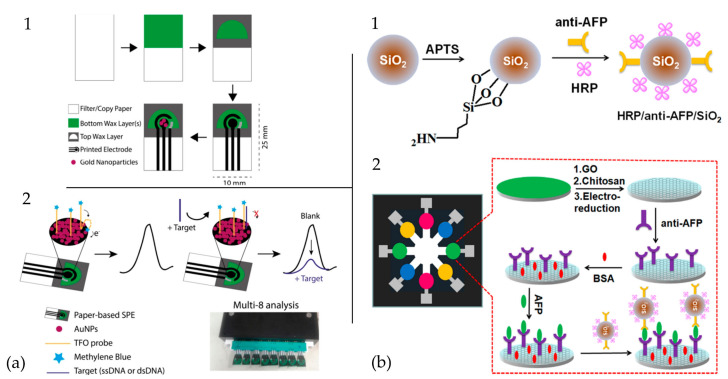
Filter paper-based biosensors (**a**) Electrochemical PAD to detect double-stranded DNA using filter paper functionalized with AuNPs. (**a1**) Components of each PAD. (**a2**) Principle of detection and prototype. Reproduced with permission from [174]. (**b**) Electrochemical PAD using filter paper modified with anti-AFP and BSA for cancer biomarkers detection. (**b1**) Nanobioprobes preparation (**b2**) Assay fabrication scheme. Reproduced with permission from [175].

**Figure 15 biosensors-11-00128-f015:**
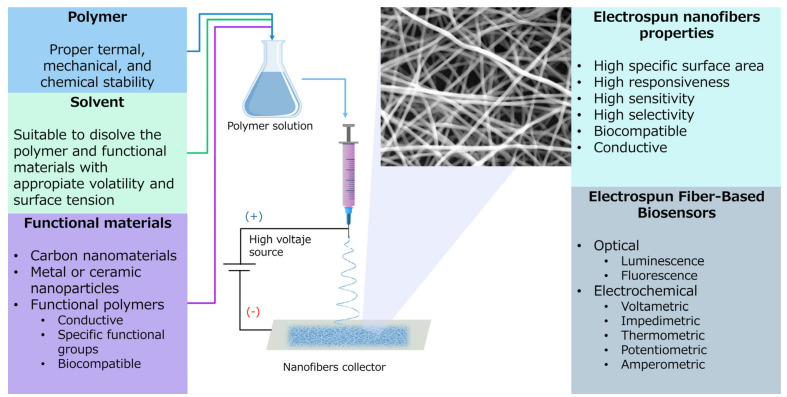
Schematic representation of the electrospinning process, properties of the materials used, the properties of the fibers, and the biosensors based on these fibers.

**Figure 16 biosensors-11-00128-f016:**
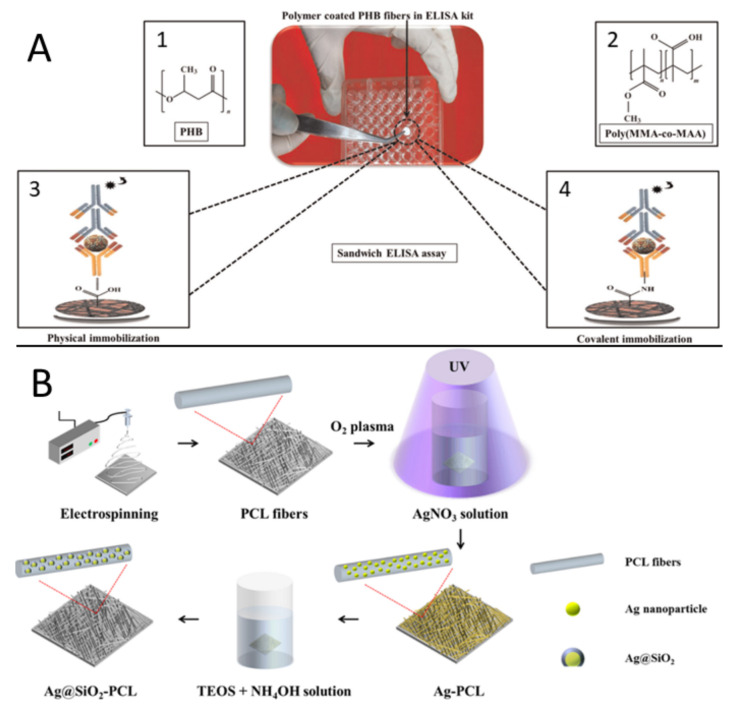
Electrospun fiber-based biosensors. (**A**) Modified ELISA bioassay with PHB nanofibers. (**a1**) and (**a2**) Are the chemical structures of PHB and poly(MMA-co-MAA), respectively. (**a3**) and (**a4**) are the physical and covalent immobilization of dengue Ab. Reproduced with permission from [136] (**B**) Shows the overall procedure for constructing a metal-enhanced fluorescence biosensor prepared using poly(caprolactone) (PCL) electrospun fibers decorated with silica-coated AgNPs. Reproduced with permission from [142].

**Figure 17 biosensors-11-00128-f017:**
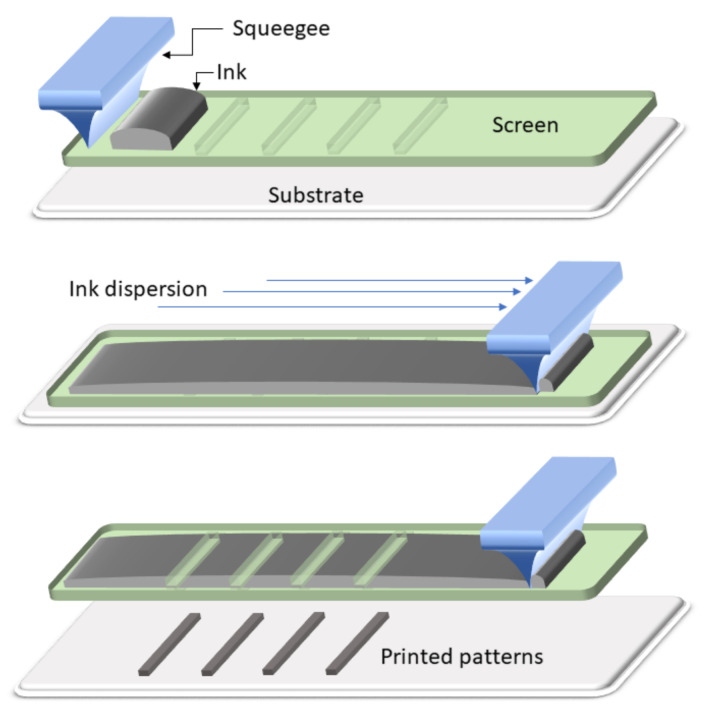
Schematic representation of the screen printing process.

**Figure 18 biosensors-11-00128-f018:**
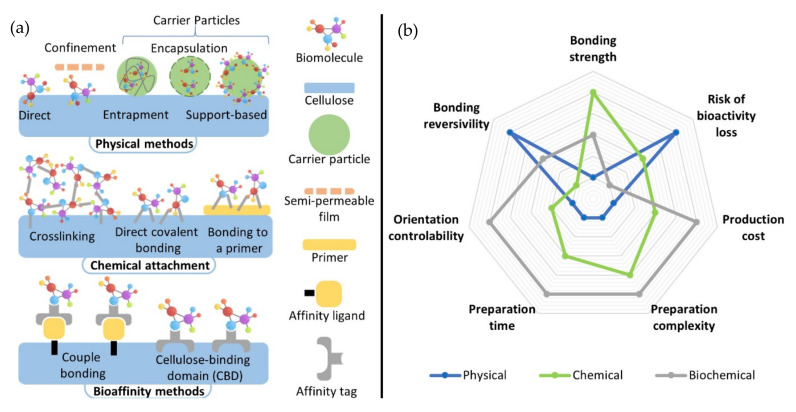
(**a**) Biomolecules immobilization method categories and their most used subcategories; (**b**) Comparison of performance and production viability.

**Figure 19 biosensors-11-00128-f019:**
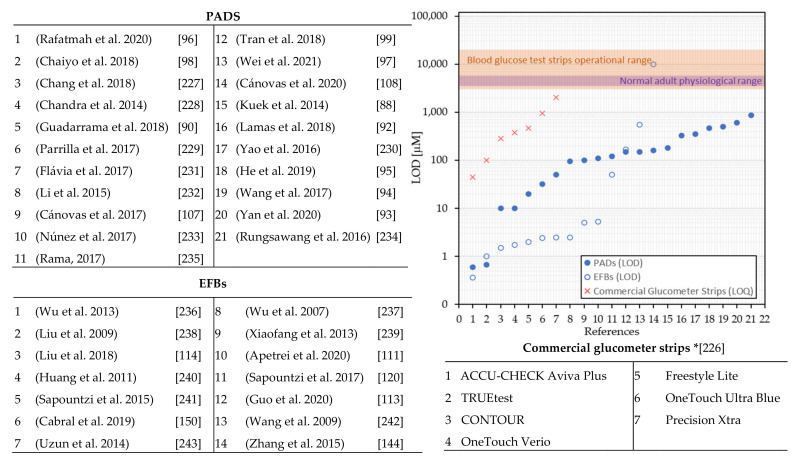
Comparison of limits of detection of different PADs and EFBs aimed to detect glucose. * Commercial glucometer strips information was adapted from Cha et al. 2017. Reprinted with permission from [226].

**Table 1 biosensors-11-00128-t001:** Classical and alternative mathematical models to describe fluid flow into paper-like materials.

	Model	Equation		Purpose	Assumptions	Definitions
**Classical model**	Lucas-Washburn (L-W)	$l\left(t\right)=\sqrt{\frac{\sigma r\cos\theta t}{2\mu }}$	(1)	To describe capillary flow in parallel cylindrical tubes	1. Constant cross-sectional area; 2. Inertial and gravitational forces are ignored; 3. Uniform pores and pore distribution; 4. Unlimited fluid reservoir volume; 5. No wicking effect due to channel boundaries; 6. Laminar flow; 7. Low-viscosity incompressible fluid; 8. No evaporation; 9. Single-phase fluid	l(t), distance traveled by the fluid σ, surface tension r, effective pore radius θ, liquid–fiber contact angle μ, fluid viscosity t, time ϕ, paper porosity h, paper thickness q_o_, evaporation rate δ, the gap between materials enclosing paper ρ, fluid density g, gravity k, experimental constant β, experimental constant θ_b_, contact angle at boundary μ_e_, effective viscosity c, constant for water
**Alternative models**	L-W modified eq. by Camplisson et al.	$l\left(t\right)=\sqrt{\frac{\gamma r\phi h\cos\theta }{4\mu q_{0}}\left(1-e^{\frac{-2q_{0}}{\phi ht}}\right)}$	(2)	To describe capillary flow in parallel cylindrical tubes, including fluid evaporation effects	1. Same as those mentioned in the L-W model with the exemption of evaporation effects.	
	L-W modified eq. by Jahanshahi et al.	$l\left(t\right)\propto \frac{2\delta }{2\delta +\phi h}{\left(\frac{\sigma^{2}t}{\rho g\mu }\right)}^{1/3}$	(3)	To describe the flow rate of fluids within paper-based microfluidic analytical devices evaluating the gravitational effects, inclination angles, and covering films.	1. Same as those mentioned in the L-W model with the exemption that the time scale associated with full penetration of the fluid into the paper is much smaller than the time scale associated with liquid rise.	
	L-W modified eq. by Hong and Kim	$l\left(t\right)=\kappa \sqrt{\left(1+\beta \frac{d}{\phi^{\frac{1}{3}}w}\frac{\cos\theta_{b}}{\cos\theta }\right)\frac{\sigma }{\mu }t}$	(4)	To describe capillary flow in parallel cylindrical tubes considering the effect due to hydrophobic barriers	1. Same as those mentioned in the L-W model: 2. Capillaries next to a hydrophobic barrier have a contact angle (θ_b_) different from those in bulk; θ_b_ > 90° to prevent imbibition through the boundary	
	L-W modified eq. by Feng et al.	$l\left(t\right)=\sqrt{\frac{\mu }{2r\mu_{e}}\left[2r+\frac{8c}{{\left(1+cos\theta \right)}^{2}}\right]}\sqrt{\frac{\sigma rcos\theta t}{2\mu }}$	(5)	To describe capillary flow in parallel cylindrical tubes considering viscosity and slippage	1. Same as those mentioned in the L-W model with the exemption of no-slip fluid-solid boundary condition and effective viscosity.	
**Classical model**	Darcy’s Law	$$Q=-\frac{\kappa WH}{\mu L}\Delta P$$	(6)	To describe flow through porous media. It can be used to characterize the flow rate in fibrous mats.	1. Incompressible fluid; 2. Viscous effects neglected; 3. Single-phase fluid; 4. Laminar flow; 5. Uniform pores and pore distribution	Q, volumetric flow rate κ, paper permeability W, channel width H, channel height L, paper length ΔP, Pressure difference V, voltage Ri, i^th^ electric resistance μ_b_, effective flow viscosity **u** = (u_x_,u_y_), average flow velocity in the porous medium S, saturation θ_i_, inclination angles of paper strip with respect to horizontal direction R_0_, initial load D, diffusive coefficient k, experimental constant β, experimental constant θ_b_, contact angle at boundary
**Alternative models**	Darcy’s Law electrical circuit analogy	$$Q=\frac{-\Delta P}{\sum_{i}^{n}\frac{\mu L_{i}}{kW_{i}H_{i}}}\overset{}{↔}I=\frac{\Delta V}{\sum_{i}^{n}R_{i}}$$	(7)	To describe a system with n-connected sections of varying geometry. The flow rate through the fluidic circuit can be modeled using an electrical circuit analogy.	1. Incompressible fluid; 2. Viscous effects neglected; 3. Single-phase fluid; 4. Laminar flow; 5. Uniform pores and pore distribution	
	Brinkman	$$-\nabla p-\frac{\mu }{k}u+\mu_{b}\nabla^{2}u=0$$	(8)	To describe fluid flow in a porous medium with high porosity.	1.Effective viscosity is assumed to be equal to the fluid viscosity; Laminar flow; 3. Incompressible fluid; 4. Single-phase fluid; 5. Uniform pores and pore distribution	
	Richards	$$\frac{\partial S}{\partial t}=\frac{\partial }{\partial x}\left[K\left(S\right)\left(\frac{\partial p\left(S\right)}{\partial x}\right)\right]+\rho gsin\theta_{i}\frac{\partial p\left(S\right)}{\partial x}$$	(9)	To describe liquid wicking behavior in thin saturated or unsaturated fibrous materials.	1. Capillary pressure and relative permeability depend on local saturation and volume of porous material. 2. Viscous effects neglected; 3. Single-phase fluid; 4. effects of inertial force and hydrostatic pressure are ignored; 5. Laminar flow	
	Elizalde et al.	$$\frac{\kappa R_{0}}{\mu }\int_{0}^{l}A\left(l∲\right)dl^{\prime }+\int_{0}^{l}\left[A\left(l∲\right)\int_{0}^{l}\frac{dx}{A\left(x\right)}\right]dl^{\prime }=Dt$$	(10)	To address fluid transport in paper with non-uniform cross-sections.	1.Inertial and gravitational forces are ignored; 2. Viscous effects neglected; 3. Single-phase fluid; 4. Laminar flow; 5. Uniform pores and pore distribution 6. Environmental effects ignored	

**Table 2 biosensors-11-00128-t002:** Literature review of some electrochemical paper-based analytical devices (PADs).

Transduction		Working Electrode			Analyte	Sample		Analytical Performance		App	Ref.
Method	Technique	Material	Fabrication	Functionalization		Source	Vol [L]	LOD [M]	Linear Range [M]		
AMP/Con	CA/EIS	PEDOT:PSS	DC	Iron Oxide NPs	CEA	Artificial serum	Not reported	Not reported	4–25 ng/mL *	CD	[82]
AMP	SWV	Cellulose	IP	GR-PANI	HPV	Synthetic HPV solution	Not reported	2.3 $\times \;10^{-3}$	10$\;\times \;10^{-9}$–200$\;\times \;10^{-9}$	CD	[83]
AMP	CV/PV	Cellulose	SP	GR	Casein	Bovine milk	Not reported	15.5$\;\times \;10^{-9}$	48.5$\;\times \;10^{-9}$–485$\;\times \;10^{-9}$	FQC	[84]
AMP	CV/DPV	Cellulose	SP/WP/DCa	AuNPs/Iron Oxide	CHIKV	Tris-EDTA buffer	3$\;\times \;10^{-6}$	0.1$\;\times \;10^{-9}$	0.001$\;\times \;10^{-6}$–100$\;\times \;10^{-6}$	CD	[85]
AMP	CV	Cellulose	SP/ES/DC	GR/PVD/PANI + SDS	Cholesterol	Human serum	Not reported	1$\;\times \;10^{-6}$	50$\;\times \;10^{-6}$–10$\;\times \;10^{-3}$	CD	[86]
AMP	DPV	Cellulose	SP	AuNPs	EGFR	Saliva	10$\;\times \;10^{-6}$	0.167$\;\times \;10^{-9}$	0.5$\;\times \;10^{-9}$–500$\;\times \;10^{-9}$	CD	[87]
AMP	CV	GR	SP	CB-PBNPs	EtOH	Beer	100$\;\times \;10^{-6}$	0.52$\;\times \;10^{-3}$	up to 10$\;\times \;10^{-3}$	FQC	[48]
AMP	CV	Cellulose	SP	-	Glucose	Sodas	5$\;\times \;10^{-6}$	0.18$\;\times \;10^{-3}$	0.5$\;\times \;10^{-3}$–5$\;\times \;10^{-3}$	FQC	[88]
AMP	CV/CA	Cellulose	SP		Glucose	Human blood	10$\;\times \;10^{-6}$	Not reported	up to 25$\;\times \;10^{-3}$	CD	[89]
POT	-	Pt	SPU	Nafion	Glucose	Orange Juice	Not reported	0.5$\;\times \;10^{-3}$	0.03$\;\times \;10^{-3}$–1.0$\;\times \;10^{-3}$	FQC	[90]
AMP	-	-	SP	-	Glucose	Human serum	0.5$\;\times \;10^{-6}$	Not reported	0–24$\;\times \;10^{-3}$	POCT	[91]
AMP	CA	Graphite	SP	-	Glucose	Soft drinks	10$\;\times \;10^{-6}$	0.33$\;\times \;10^{-3}$	0.5$\;\times \;10^{-3}$–50$\;\times \;10^{-3}$	FQC	[92]
AMP	CA	Au/Carbon	ED	MSA/EDC/NHS	Glucose	Artificial serum	Not reported	0.6$\;\times \;10^{-6}$	2$\;\times \;10^{-6}$–21.97$\;\times \;10^{-3}$	CD	[93]
AMP	-	Cu/RGO	PD	-	Glucose	Artificial serum	0.1$\;\times \;10^{-3}$	0.5$\;\times \;10^{-6}$	2 $\;\times \;10^{-6}$–2$\;\times \;10^{-3}/$ 2$\;\times \;10^{-3}$–13$\;\times \;10^{-3}$	CD	[94]
AMP	CA	C/PE	SP		Glucose	Glucose solution	16$\;\times \;10^{-6}$	470$\;\times \;10^{-6}$	0–16$\;\times \;10^{-3}$	CD	[95]
AMP	CA	Au/Cellulose	ED	Nano-Dendritic Au	Glucose	Glucose solution	Not reported	0.6$\;\times \;10^{-6}$	10$\;\times \;10^{-6}$–15$\;\times \;10^{-3}$	CD	[96]
AMP	-	Co-MOF/C cloth/filter paper		-	Glucose	Glucose solution	Not reported	0.15$\;\times \;10^{-3}$	0.8$\;\times \;10^{-3}$–16$\;\times \;10^{-3}$	CD	[97]
AMP	CA	CoPc/GR/IL/C/filter paper	SP		Glucose	Human serum/Honey	50$\;\times \;10^{-6}$	0.67$\;\times \;10^{-6}$	0.01$\;\times \;10^{-3}$–1.3$\;\times \;10^{-3}$/1.3$\;\times \;10^{-3}$	CD	[98]
AMP	CV	Au NPs/SWCNTs/NC	ED/WP	-	Glucose	Glucose solution	Not reported	148$\;\times \;10^{-6}$	0.5$\;\times \;10^{-3}$–10$\;\times \;10^{-3}$	POCT	[99]
AMP	CV	Cellulose	SP	-	H2O2	Lens cleaning sol.	5$\;\times \;10^{-6}$	4.1$\;\times \;10^{-6}$	0.02$\;\times \;10^{-3}$–0.5$\;\times \;10^{-3}$	CD/Env.	[100]
Con	LSV	Ag	Brush painting	-	HSA	HSA-PB/BSA-PB sol.	20$\;\times \;10^{-6}$	1$\;\times \;10^{-12}$	0.015$\;\times \;10^{-9}$–9.43$\;\times \;10^{-9}$	CD	[101]
AMP/IMP	CV/EIS	Graphite	SP	-	L-Tyrosine	HB plasma	3$\;\times \;10^{-6}$	0.02$\;\times \;10^{-6}$	50$\;\times \;10^{-9}$–600$\;\times \;10^{-6}$	CD	[102]
AMP	CV	Cellulose	-	PBNPs + Cu	MeBut	Candies/Essences	Not reported	0.8$\;\times \;10^{-3}$	0.25$\;\times \;10^{-6}$–30$\;\times \;10^{-6}$	FQC	[30]
AMP	CV/CA	Cellulose	SP	CB-PBNPs	Nerve agents	Paraoxon	5$\;\times \;10^{-6}$	3 μg/L *	0–25 μg/L *	EM	[103]
AMP	CV/DPV	Cellulose	SP	Au NRs	Ovalbumin	-	5$\;\times \;10^{-6}$	19$\;\times \;10^{-15}$	22$\;\times \;10^{-15}$–22$\;\times \;10^{-9}$	CD	[104]
AMP	CA	Graphite	SP	CB + PBNPs	Atrazine	River water	5$\;\times \;10^{-6}$	9.3$\;\times \;10^{-9}$	93$\;\times \;10^{-9}$–464$\;\times \;10^{-9}$	EM.	[47]
IMP	-	Ag	DC	-	PSA	PSA + PB solutions	Not reported	39$\;\times \;10^{-12}$	0–17$\;\times \;10^{-9}$	CD	[105]
AMP	-	GR	SP	CB-PBNPs	Sulfur mustard	Mustard agent solutions	1.5$\;\times \;10^{-6}$	1$\;\times \;10^{-3}$	0–6$\;\times \;10^{-3}$	POCT	[35]
AMP	SWV	Cellulose	SP	GR quantum dots	UA/CREAT	Human urine	Not reported	3.7 1$\;\times \;10^{-6}$	10$\;\times \;10^{-9}$–3$\;\times \;10^{-6}$	CD	[106]
POT	-	Pt/filter paper	SPU	Nafion	Glucose	Artificial serum	25$\;\times \;10^{-6}$	0.1$\;\times \;10^{-3}$	0.3$\;\times \;10^{-3}$–3$\;\times \;10^{-3}$	POCT	[107]
POT	-	Pt/filter paper	SPU	Aquivion	Glucose	Artificial serum	25$\;\times \;10^{-6}$	0.16$\;\times \;10^{-3}$	0.5$\;\times \;10^{-3}$–10$\;\times \;10^{-3}$	POCT	[108]

* Data reproduced from the original work. Not enough information for its conversion into concentration molar units.

**Table 3 biosensors-11-00128-t003:** Literature review of some electrochemical electrospun fiber-based biosensors (EFBs).

Transduction		Electrospun Mats				Analyte	Recognition	Sample	Analytical Performance		App.	Ref.
Method	Technique	Collector	ES Solution	Function	Functionalization Post-ES				LOD [M]	Linear Range [M]		
AMP	CV	-	PVA/PAMAM-Mt/GC	Substrate	GA	Glucose	E: PyOx	Soft drink cola	0.7$\;\times \;10^{-3}$	5$\;\times \;10^{-6}$–0.25$\;\times \;10^{-3}$	FQC	[109]
AMP	CA	FTO	PVP/Chi/rGO	Substrate	GA	17α-EE	E: Laccase	Human urine	0.15$\;\times \;10^{-12}$	Not reported	CD	[110]
AMP	-	Pt	PAN/Mt	Substrate	DDAC	Glucose	E: GOx	Fruit juices	2.4$\;\times \;10^{-3}$	1$0\;\times \;10^{-6}$–2.45$\;\times \;10^{-3}$2.45$\;\times \;10^{-3}$–15$\;\times \;10^{-3}$	FCQ	[111]
AMP	CV	ITO	PAN/AgNO3	Coating	EDC/NHS	Triglyceride	E: Lip-GLDH	Artificial sample	0.6$\;\times \;10^{-9}$	2.3$\;\times \;10^{-3}$–$\;\times \;10^{-3}$	CD	[112]
AMP	CV	GC	PAN	Precursor	Pyrolysis: NiCo_2_S_4_/EGF	Glucose	-	Glucose solution	0.167$\;\times \;10^{-3}$	0.5$\;\times \;10^{-6}$–3.571$\;\times \;10^{-3}$	POCT	[113]
AMP	CV	GC	PAN	Precursor	Carbonization: NiCo_2_O_4_/ECF	Glucose	-	Glucose solution	1.5$\;\times \;10^{-3}$	$5\;\times \;10^{-3}$–19.175$\;\times \;10^{-3}$	POCT	[114]
AMP/IMP	CV/EIS	GC	PAN/SnO_2_	Coating	MPA/EDC-NHS	Atrazine	I: anti-atrazine Ab	Spiked water	0.9$\;\times \;10^{-21}$	1$\;\times \;10^{-21}$–1$\;\times \;10^{-6}$	EM	[115]
IMP	ElS	-	PEDOT/NBR	Substrate	ON probes/PAA brushes	NHL gene	DNA	Artificial solution	1$\;\times \;10^{-18}$	1$\;\times \;10^{-18}$ to 100$\;\times \;10^{-12}$	CD	[116]
IMP	EIS	-	PAN	Coating	-	Zearalenone	-	Artificial food	1.66$\;\times \;10^{-9}$	5$\;\times \;10^{-9}$–30$\;\times \;10^{-9}$ 60 $\;\times \;10^{-9}$ to 100 $\;\times \;10^{-9}$	-	[117]
IMP	EIS	-	CAc	Substrate	ZIF-8/MWCNTs/Au	Glucose	E: GOx	Synthetic sample	5.347$\;\times \;10^{-3}$	1–10$\;\times \;10^{-3}$	CD	[118]
IMP	EIS	FTO	PA6/PPy	Coating	ZnO NPs	Urea	E: Urease	Milk	1.8$\;\times \;10^{-6}$	17$\;\times \;10^{-6}$–42$\;\times \;10^{-3}$	FQC	[119]
IMP	EIS	Au	PVA/PEI	Coating	Au NPs	Glucose	E: GOx	Synthetic sample	0.9$\;\times \;10^{-6}$	10$\;\times \;10^{-6}$–200$\;\times \;10^{-6}$	CD	[120]
IMP	EIS	CPE	PVA/Honey	Coating	Au NPs/MWCNTs	CEA	I: Anti-CEA	Clinical serum	0.5 $\times \;10^{-12}$	2.2 $\times \;10^{-12}$–694$\;\times \;10^{-12}$	CD	[121]
IMP	CR	Si glass	PANi/PEO	Coating	-	DENVCP	DNA probe	Blood serum	1.9 $\times \;10^{-15}$	10 $\times \;10^{-3-15}$–1 $\times \;10^{-6}$	CD	[122]
POT	-	-	PMMA	Substrate	Ca2+ Ionophores/Nafion/Au	Calcium ions	Ionophores	Artificial sweat	14$\;\times \;10^{-9}$	1$\;\times \;10^{-3}$	CD	[123]

* Data reproduced from the original work. Not enough information for its conversion into concentration molar units.

**Table 4 biosensors-11-00128-t004:** Literature review of some optic paper-based analytical devices (PADs).

Transduction		Analyte	Sample		Recognition Element	Analytical Performance		Response Time [min]	App.	Ref.
Method	Principle		Source	Volume [μL]	Type	LOD [M]	Linear Range [M]			
Spectroscopy	CL	PSA	HS	5	I: Anti-PSA Ab	26$\;\times \;10^{-15}$	33$\;\times \;10^{-15}$–0.67$\;\times \;10^{-9}$	10	CD	[45]
Colorimetry	-	CEA	HS	5	I: Anti-CEA Ab	14$\;\times \;10^{-9}$	28$\;\times \;10^{-15}$–167$\;\times \;10^{-15}$	120	CD	[46]
Colorimetry	-	*E. coli*/*L. monocytogenes*/*S. aureus*	Synthetic sample	40	DNA: Biotinylate capture probes	1 pg/μL *	1 ng/μL–1 pg/μL *	40–50	FQC/CD	[133]
Colorimetry	-	Immunoglobulins	Bovine serum albumin	5	I: Biotinylated antimouse IgG Ab	2$\;\times \;10^{-6}$	Not reported	24	CD	[134]
Colorimetry	-	Alkaline phosphatase	Milk	-	I: Anti-ALP Ab	0.87 U/mL *	10–1000 U/mL *	13	FQC	[135]
Colorimetry	-	Glucose	Artificial Urine	5	E: GOx+HRP	Not reported	0–2$\;\times \;10^{-9}$	30	CD	[43]
Colorimetry	-	Dengue	Synthetic sample	100	I: Anti-Dengue Ab	8$\;\times \;10^{-6}$ p.f.u/mL *	Not reported	-	CD	[136]
Colorimetry	-	Paromomycin sulfate/Tetracycline/Hydrochloride/chloramphenicol/erythromycin	Water	2	E: b-galactosidase	0.5, 2.1, 0.8 6.1 μg/mL *	Not reported	120–1440	Env.	[137]
Colorimetry	-	Glucose	Human Saliva	50	E: GOx	1.2$\;\times \;10^{-3}$	0.5$\;\times \;10^{-6}$–75	0.75	CD	[138]
Spectroscopy	Fluorescence	Phakopsora Pachyrhizi	Soybean	2	I: Anti-Phakopsora Pachyrhizi Ab	2.2 ng/mL	0.0032–3.2 μg/mL *	60	FQC	[139]
Spectroscopy	Fluorescence	Gold ions	Human Urine	1	WC: Cupriavidus metallidurans	110$\;\times \;10^{-9}$	Not reported	-	CD	[140]
-	Fluorescence	AFP	HS	2.5	I: Anti-AFP Ab	6$\;\times \;10^{-15}$	14.3$\;\times \;10^{-15}$–12.9$\;\times \;10^{-12}$	60	CD	[141]
Spectroscopy	SPR	Bovine haptoglobin	Bovine serum	10	I: Anti-haptoglobin Ab	28 μg/mL *	0.01–0.9 mg/mL *	5	CD	[142]

* Data reproduced from the original work. Not enough information for its conversion into concentration molar units.

**Table 5 biosensors-11-00128-t005:** Literature review of some optical electrospun fiber-based biosensors (EFBs).

Transduction		Electrospun Mats				Analytical Performance						
Method	Technique	Collector	ES Solution	Function	Functionalization Post-ES	Analyte	Recognition	Sample	LOD [M]	Linear Range [M]	App.	Ref.
FLU	MEF	-	PCL	Substrate	Ag@SiO_2_/PCL	IgG	Immuno-based	Artificial solution	$10\;\times \;10^{-9}$	Not reported	CD	[143]
FLU	SM	Al foil	PAN/pVDB	Substrate	Boronic acid	*S. aureus*/*E. coli*	-	Beef-based nutrient broth	Not reported	Not reported	CD	[144]
FLU	SM	ITO	PVA/GQD	Coating	-	Glucose	GOx	Glucose solution	10$\;\times \;10^{-3}$	0.25$\;\times \;10^{-3}$–24$\;\times \;10^{-3}$	CD	[145]
SPE	SPM	-	Chi/PVA	Substrate	Guaiacol	Time/Temperature	Laccase	4 °C Environment	Not reported	1–38 days *	FCQ	[146]
SPE	SERS	-	PCL	Substrate	4-MB/Au NPs/Ag NPs	PSA	Anti-PSA	Artificial solution	0.03$\;\times \;10^{-12}$	Not reported	CD	[147]
SPE	COL	-	PVA	Precursor	Red Cabbage Pigment	pH	Pigment	Fruit surfaces	Not reported	2–12 pH *	FQC	[148]
SPE	COL	-	PHBV	Substrate	Nafion/BSA/GA	Paraoxon	AChE	Artificial solution	36.3$\;\times \;10^{-12}$	36.3$\;\times \;10^{-12}$–0.2$\;\times \;10^{-9}$	EM	[149]
SPE	UV-vis	Optical fiber	PPO	Coating	-	Ammonia	-	Volatile Vapor	5.87$\;\times \;10^{-9}$	Not reported	CD	[150]
-	SPM	ITO	PVA/GQD	Coating/Substrate	graphene QD	Glucose	GOx	Artificial solution	12$\;\times \;10^{-3}$	1$\;\times \;10^{-3}$–10$\;\times \;10^{-3}$	CD	[151]
SPE	COL/SPM	Al foil	PVA/anthocyanin	Substrate	Glutaraldehyde	Ferric ions	Anthocyanin	Water	17.9$\;\times \;10^{-6}$	17.9$\;\times \;10^{-6}$–6.3$\;\times \;10^{-3}$	POCT	[152]

* Units as reported by authors.

## Data Availability

No new data were created or analyzed in this study. Data sharing is not applicable to this article.

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
