# Peer review of "Paper and Other Fibrous Materials—A Complete Platform for Biosensing Applications"

_biosensors, 2021, doi:10.3390/bios11050128_

Round 1
Reviewer 1 Report
The review article "Paper and other fibrous materials: A complete platform for bio-2 sensing applications" perfectly suits the Journal of Biosensors. This review is very complete and it is innovative. This work distinguishes itself from other reviews of paper sensing platforms since it covers all detection methods of biosensing, bringing equations describing fluid flow into paper-like materials.
I have no concerns about publishing the paper in the present form, due to its high standard quality, congratulation to the authors for this relevant review.
Author Response
We really appreciate your kind words.
P.S. Since there are no comments to address, no documents were uploaded with responses.
Reviewer 2 Report
Overall, the authors provide a good overview on the topic, which is helpful, in particular, for persons new in the field. However, the structure could still be improved.
The abstract needs significant improvement to convey the core message to the readers.
The comparison and integration of electrospun materials is not well described. Electrospun material and the method of electrospinning is mentioned for the first time on page 3, line 99, while in the following text, paper and fibrous materials are not clearly differentiated, which is confusing.
The locations of Tables 2 and 3 is confusing. It would be better to provide more text prior to these talbes. Moreover, the explanation and introduction of abbreviations mentioned in tables 2 and 3 is insufficient.
Page 10, line 269: Non-suitable citation - the cited paper is also a review. Besides, the cited paper mentions electrochemical sensing but does not give any details on the different electrochemical methodologies, as suggested in the text.
Until the very end of the review it remains unclear if the authors would like to compare natural paper and electrospun fibers. Only after reading “Conclusions and Future Perspectives“ it becomes clear that it is a comparison.
Reviewer 3 Report
The present review entitled “Paper and other fibrous materials: A complete platform for bio-sensing applications” is an interesting work that summarizes and compares the fabrication trends in paper-based analytical devices (PADs) covering different types of immobilization and functionalization together with the development of microfluidics pathways. The authors also reviewed classical and alternative models of adsorption as well as transduction mechanisms for the readout applied in different scenarios. Therefore, this review could result in a future guide for the selection of materials, bio-immobilization approaches, fabrication techniques, mathematical models, and readout methodologies to develop complete and robust biosensing platforms. All general schematic figures created by the authors summarize in a very well manner the concepts, ideas and fabrication methods, being helpful for the reader. Even though the present review contains useful information for the field, some extra examples should be included and the critical discussion of the achievements and limitations (strengths and weaknesses) of some cited examples need to be improved. I highly recommend providing these suggestions which will lead to a more complete and relevant manuscript.
Please address the following points before accepting this manuscript for its publication in Biosensors Journal:
- Please, carefully check the proper order of subject, verb and predicate, that grammatical structure must be followed throughout the abstract and during the manuscript in general.
- Line 27-28: please replace geographical by geographic or demography by demographic, unify accordingly.
- Line 36-37: “In recent years, paper-based analytical devices PADs have transformed health, and environmental applications for POC approaches.” Please include relevant references herein (examples from George Whitesides group should be added since he is considered one of the pioneers of the µPADs).
- Lines 57-59: please try to be more specific and clarify the “etc” of each factor.
- The schematic representations of figure 1 results in a useful and clear image that summarizes different designs used in the literature. I recommend aligning the subtitles, double-check the size and fonts and keep them always in the same position: on top of the corresponding design or below.
- Line 74: “…it is needed the specific material selection (…)” maybe rephrase this sentence to: “several factors should be considered such as: the specific material selection, the size, and arrangement…
- Line 142: Understating the fluid dynamics… I believe it was a mistake, the authors wanted to minimize the importance of the fluid dynamics or do they want to highlight exactly the opposite? Maybe the authors wanted to say “Understanding”? Please rephrase accordingly.
- Line 226: Section 3 is called Paper-based Analytical Devices (PADs) and all its content is focused on PADs. However, in the first sentence electrospun fiber-based biosensors (EFBs) are also mentioned. I think more information about the EFBs could enrich the review, if possible. For instance, an image similar to image 3 showing the typical format of EFBs and some specific examples inside sections 3.1 and 3.2 (including EFBs also in the titles) or create the corresponding sections for EFBs depending on their readout.
- Line 236: in this part of the review where the lateral flow assays are mentioned, I think articles with high importance such as the Nature Protocols 15 (12), 3788-3816 recently published by Arben Merkoçi should be added somewhere.
- Please carefully check the units in tables 2 and 3, there are some cases missing and try to unify the units. Please choose if you want to show them in molar range or if you prefer ng/mL, for instance, this should be the same throughout the entire manuscript. Even though the potentiometric readout is less common, the authors could consider including more examples in both tables, for table 2 (DOI: 10.1039/C7LC00339K and https://doi.org/10.1016/j.bios.2020.112302) and for table 3, see my next comment, please.
- Table 3, there is a mistake in the title “electrpun” and there is an interesting work performed with carbon fibers to develop another example of a potentiometric sensor that could be included in this table and throughout the discussions in the text: https://doi.org/10.1002/elan.201600070
- Line 337: “A major associated disadvantage is the requirement of auxiliary equipment…” the already mentioned example (https://doi.org/10.1016/j.bios.2020.112302) could be included since I found this example very useful for your review because it is a paper-based microfluidic potentiometric sensor for the detection of glucose in whole blood with the easy integration of a miniaturized potentiostat able to send the readout signal to a mobile phone or tablet through its specific and user-friendly app.
- Regarding optical microfluidic PADs (section 3.1), the authors should consider including some of the relevant articles published in this particular field by the group of Prof. Capitan-Vallvey, such as: https://doi.org/10.1021/ac5019205 ; https://doi.org/10.1016/j.snb.2021.129506 ; https://doi.org/10.1016/j.talanta.2020.121108 among others.
- I believe the addition of this interesting work developed for the detection of creatinine https://doi.org/10.1016/j.aca.2017.11.026 as well the aforementioned examples will contribute to the creation of a more complete review comprising a wide variety of examples.
- Figure 10. Part (A) is missing in the caption and the resolution of the figure should be improved, now it is a bit blurry.
- Could the authors elaborate a bit more in the explanation of Guo and co-workers in the paragraph starting in line 437? Which are the disadvantages of that specific example? Normally when a specific article is explained, this explanation should include the strengths and weaknesses of that work, thus, future readers can identify what is needed and not achieve yet. There are some examples commented along with the text in the same way, please try to improve them by introducing the strengths and weaknesses from a critic's point of view.
- Line 502: figure 12 is inserted in between text.
- Please double-check that all figures taken from other articles include in the caption “Reproduced with permission from [XX]”, for example in Figure 13 and 14?
- Line 583: “Gold nanoparticles” is more commonly found in literature as “AuNP” instead of “GNP”. Therefore, I would recommend using “AuNP” since I think it will be easier to follow for the readers in general.
- Line 588: several targets are mentioned (“…target analytes such as trichloro-pyridino, olaquindox residues, DNA, thrombin, carbofuran, glucose, and cancer cells”) but only a couple of references are cited at the end of the sentence. Could the authors specify a reference for each case and placed it close to the corresponding one?
- Line 591: “…selectivity. Examples: Cerium oxide, copper oxide. Nickel oxide, zinc oxide, titanium oxide, among others.” Please, rephrase the sentence and integrate the examples. For example: …selectivity, such as: cerium oxide, copper oxide, nickel oxide, zinc oxide or titanium oxide, among others.
- Line 615: double check and correct the repeated references showed herein: “[161][162][162][162][162][162][162][162][163][163].”
- Line 641: the acronyms only should be introduced in the text if they are going to be used in other parts of the manuscript. “S” for sputtering is never used again. Please, double-check this issue throughout the entire review with the rest of the acronyms and correct accordingly.
- Line 696: “uPAD”. Please, introduce the symbol to write it in the proper way: µ
- During the entire manuscript, sometimes the authors cite other authors with the last name of the first author et al. (e.g. Kumar et al.), other times the first letter of the name followed by the last name et al. (e.g. S. Kumar et al.) and another with the name and last name (e.g. Todd Hoare et al.). The proper and most usual way is the first one, please double-check with the format required by this Journal and most importantly, unify all citations along with the text so the same format is kept during the entire document.
- Line 767: “resulting in microchannels wider than expected” are the authors sure about this? Wider or narrower than expected?
- Line 843: definitely the aforementioned work published by Canovas et al. (DOI: 10.1039/C7LC00339K (Paper) Lab Chip, 2017, 17, 2500-2507) as well as the work recently provided by the same authors (https://doi.org/10.1016/j.bios.2020.112302) should be included as well for the discussion of this interesting figure 17 since both articles are focused in the potentiometric detection of diluted blood (in the first case) and undiluted blood (in the most recent case) by tuning the analytical response by controlling the mixed potential.
- The reference 205 listed as “Parrilla, M.; Cánovas, R.; Andrade, F.J. Biosensors and Bioelectronics Paper-Based Enzymatic Electrode with Enhanced Potentiometric Response for Monitoring Glucose in Biological Fluids. Biosens. Bioelectron. 2017, 90, 110–116.” is not mentioned or cited in the entire text. And in Figure 17, the name of the first author has been changed, please double-check and correct accordingly.
Round 2
Reviewer 2 Report
The manuscript has been improved considerably.
The authors should carefully check for literature on electrochmical LFDs. Relevant references seem to be missing in this study, e.g. https://doi.org/10.1039/C3AN02328A and 10.1016/j.bios.2020.112015.
Author Response
We appreciate the comments and suggestions of Reviewer 2 in this 2nd round of revisions. Please see the attachment for details.
